# Attention Dynamics in Spatial–Temporal Contexts

**DOI:** 10.3390/bs15050599

**Published:** 2025-04-30

**Authors:** Yuying Wang, Xuemin Zhang, Eva Van den Bussche

**Affiliations:** 1Beijing Key Laboratory of Applied Experimental Psychology, National Demonstration Center for Experimental Psychology Education, Faculty of Psychology, Beijing Normal University, Beijing 100875, China; 2Brain and Cognition, Faculty of Psychology and Educational Sciences, KU Leuven, 3000 Leuven, Belgium; 3State Key Laboratory of Cognitive Neuroscience and Learning, Beijing Normal University, Beijing 100875, China

**Keywords:** attention, psychological timeline, short-term memory, Gestalt perceptual organization principles

## Abstract

This study systematically explored the impact of the *spatial metaphor of time* on attentional bias using visual order stimuli. Three experiments were conducted. Experiment 1, comprising Experiments 1a and 1b, investigated how the *spatial metaphor of time* shaped attentional bias across five disorder levels. Experiment 1a examined horizontal orientations, while Experiment 1b focused on vertical orientations. Experiment 2 compared attentional biases toward stimuli with the same disorder level in both orientations. The key distinction between the stimuli was that one represented short-term memory, while the other did not involve memory. Building on the findings of Experiment 2, Experiment 3 compared attentional biases between fully ordered structures (intact Gestalt structures) as non-memory representations and partially disordered structures in short-term memory. The results revealed a significant preference for future-related information, particularly on the right side in horizontal orientations. Short-term memory representations enhanced attentional attraction and triggered inhibition of return (IOR), while fully ordered structures attracted attention as effectively as partially disordered structures, thereby neutralizing attentional biases. Overall, this study contributes to a deeper understanding of the cognitive mechanisms underlying visual attention and the influence of temporal metaphors.

## 1. Introduction

The human brain integrates spatial and temporal dimensions through complex cognitive mechanisms, often associating specific spatial positions with temporal sequences. This process, known as serial order, involves arranging events or items in a sequence ([32]) and is fundamental to cognitive processing. It reflects the *spatial metaphor of time* ([11]), illustrating the brain’s ability to synthesize sensory inputs into coherent temporal representations ([46]; [76]). Studies have shown that individuals from left-to-right reading cultures, such as English and Spanish speakers, are more likely to perceive stimuli on the left as occurring earlier than those on the right ([47]; [54]). Moreover, the concepts of “past/future” and “before/after” are intrinsically linked in spatial–temporal representations. While “before/after” denotes a sequential relationship, “past/future” represents a more abstract, directional understanding of time. In left-to-right reading cultures, these concepts converge, with “before” and “past” typically associated with leftward positions and “after” and “future” linked to rightward spatial locations. This conceptual alignment underscores the cognitive mechanism by which temporal sequences are mapped onto spatial dimensions ([57]). While this metaphor is most evident in left/right orientations, it also extends to other dimensions, such as front/back and top/bottom, though findings on the vertical dimension remain inconclusive ([5]; [11]; [29]; [39]; [55]). Neuroimaging research has revealed that maintaining order information in working memory engages specific cortical networks, particularly the prefrontal cortex and parietal lobes, highlighting the neural basis of these cognitive processes ([42]; [69]). Recent work by [41] ([41]) further demonstrated a gradual shift of attention from past to future locations, mapping temporal progression within the *spatial metaphor of time*. These findings underscore the importance of serial order processing in spatial dimensions, suggesting that the *spatial metaphor of time* may significantly influence attentional mechanisms.

Attention, as a sophisticated cognitive process, integrates and prioritizes information based on various factors. Among these, the basic features of objects, such as color and shape, play a foundational role in guiding attention ([9]; [22]; [24]; [77]). Beyond these individual attributes, the structural organization facilitated by Gestalt perceptual grouping is equally critical. Gestalt perceptual grouping refers to the principles by which humans tend to organize visual elements into wholes. These principles include proximity, similarity, and continuity, which help us quickly recognize and understand information in complex visual environments ([3]; [71]). Gestalt principles enhance the efficiency of perceptual organization, enabling attention to optimize information processing and underscoring their importance in visual cognition ([4]; [21]; [25]; [35], [36]; [67]). The structural features of Gestalt combined with attributes like color and shape can form object perception, while object-based attention further enhances visual search performance by significantly improving both efficiency and accuracy ([52]). This advantage arises from the ability of unified entities to capture and maintain attention more effectively ([75]). Entities with greater salience and stability tend to make the attentional focus more resilient to environmental disruptions ([8]; [10]; [40]; [77]). This resilience can be attributed to the fact that complete and unified Gestalt structures are more likely to be perceived as wholes compared to incomplete structures containing disordered parts ([70]). To enhance our understanding of how individuals process information across different spatial locations, it is worthwhile to investigate how the integrity of object perception influences attention allocation rather than focusing solely on the spatial positioning of objects. In this context, it is vital to explore the influence of the *spatial metaphor of time* on object-based attentional bias and the role of Gestalt perceptual grouping in shaping attentional mechanisms. For example, it is important to investigate whether stimuli presented in the right visual field, which are associated with future-oriented concepts, would attract more attention compared to those presented in the left visual field. Additionally, the question arises as to whether varying levels of disorder in stimuli would lead to differences in attention and whether these differences would manifest differently between different visual fields.

Memory is intricately linked to attention ([2]; [18]; [37]; [45]). The effective allocation of attention enhances the encoding and maintenance of spatial–temporal cues in working memory, facilitating tasks reliant on visual cognition and spatial processing ([14]; [17]; [23]; [34]; [51]). Moreover, research has demonstrated that working memory can actively guide attention allocation, particularly when external information aligns with stored memory representations. In such cases, attention is directed preferentially toward these consistent elements ([7]; [20]; [60]). [73] ([73]) further revealed that when visual sensory memory representations are relevant to the current task, attention is preferentially allocated to those representations. Given that working memory and visual short-term memory share largely overlapping underlying capacity limitations ([1]; [13]), visual short-term memory might consistently affect attention in a manner comparable to both working memory and sensory memory. Given these insights, it becomes essential to conduct experiments that integrate short-term memory to investigate how objects organized within memory representations differ in attracting attention compared to those not represented in memory. Furthermore, considering that intact Gestalt structures are more likely to be perceived as wholes compared to non-intact Gestalt structures ([70]), it is also pertinent to explore whether intact Gestalt structures attract attention more effectively than incomplete ones or whether they exhibit similar advantages in attention capture as non-intact Gestalt structures represented in memory do over unrepresented incomplete structures.

In summary, the present study aimed to explore the impact of the *spatial metaphor of time* on attentional bias while further investigating the roles of visual short-term memory and Gestalt perceptual grouping in this effect. Using a spatial cueing paradigm ([49]; [52]), we manipulated visual order by using images with varying levels of “disorder” as cues. These levels (0D, <1/4D, 1/4D, 1/2AD, 1/2ND) were determined by the proportion of disrupted perceptual structure within a four-quartered image, with “D” representing disorder. The disorder levels were established by dividing each image into four quarters and introducing disorder in specified portions: complete order (0D), less than one-quarter disorder (<1/4D), one-quarter disorder (1/4D), two adjacent quarters disorder (1/2AD, with “A” representing adjacent), and two non-adjacent quarters disorder (1/2ND, with “N” representing non-adjacent). This systematic variation in visual complexity allowed us to assess its influence on attentional bias (as shown in Figure 1). This manipulation of visual order was employed across three experiments. Experiments 1a and 1b investigated attentional bias under varying levels of disorder and orientations, while Experiment 2 examined the effect of the short-term memory representation on subsequent attentional bias. Experiment 3 further tested whether short-term memory representations elicited similar effects on attention as fully ordered (intact Gestalt) structures.

The primary hypotheses of this study are as follows: First, we hypothesized that object-based attentional bias is shaped by spatial position, reflecting the dynamic nature of attentional shifts across the *spatial metaphor of time*. Specifically, we hypothesize that individuals will exhibit an attentional bias toward stimuli on the right in the horizontal orientation, which is consistent with the future-oriented spatial metaphor ([41]; [47]; [54]). Therefore, we expect that participants will show faster reaction times and higher accuracy rates for these stimuli compared to those on the left. Similarly, we hypothesize that in the vertical orientation, stimuli located downward will attract more attention than those located upward, potentially reflecting a similar temporal association with future-oriented positions. Thus, we expect participants to demonstrate faster responses and improved accuracy for downward stimuli compared to upward stimuli. Second, we hypothesize that stimuli represented in short-term memory will attract more attention compared to non-represented structures. We anticipate that in the experiments, participants will show significantly faster reaction times and higher accuracy rates for stimuli that are represented in short-term memory. Finally, we hypothesize that competition for attentional capture may occur between fully ordered structures (0D) and structures with disorder (<1/4D, 1/4D, 1/2AD, 1/2ND) represented in short-term memory, and this leads to no significant bias toward either type. We expect that in such competitive situations, participants’ attention may be divided between the two types of stimuli, resulting in no significant difference in reaction times or accuracy. This outcome could imply that attentional capture is influenced by both types of stimuli. Additionally, this result might suggest that both types are likely more effective in capturing attention than structures containing disorder in non-short-term memory representations.

## 2. Experiment 1

### 2.1. Method

#### 2.1.1. Participants

The minimal sample size was determined by a priori power analysis using G*Power software, version 3.1.9.7 ([19]). A power analysis was conducted to determine the required sample size for the study. It was estimated that a minimum of 14 participants would be necessary to achieve a statistical power of 80% for detecting medium effect sizes (*f* = 0.25) at a significance level of 5%. This sample size was specifically calculated to assess the 2 (image position) × 5 (disorder level) within-subject interaction in Experiments 1a and 1b. In Experiment 1a, 20 college students aged 18 to 24 years (5 males, *M*_age_ = 21.00 years, *SD* = 1.55) participated, 19 right-handed and one left-handed. In Experiment 1b, a different group of 20 college students aged 19 to 24 years participated (4 males, *M*_age_ = 21.45 years, *SD* = 1.60), all right-handed. The Ethics Committee of the Psychology Faculty at Beijing Normal University approved this study ((BNU202310050144). All participants signed the informed consent and received payment for their involvement.

#### 2.1.2. Apparatus and Stimuli

The desktop computer, connected to a 27-inch color monitor with a refresh rate of 60 Hz and a resolution of 2560 × 1440 pixels (57.08° × 32.10°), displayed the stimuli and collected the data. The experiment was administered using PsychoPy (v2021.2.3). Participants were permitted to adjust their viewing distance slightly, maintaining an approximate distance of 60 cm for optimal comfort.

The stimuli used in the experiments were OCTA-generated images developed by [63] ([63], [64]). Forty images were used, divided into five sets of eight images each. These sets were categorized based on the proportion of disorder present in the images. The disorder levels were determined by dividing each image into four quarters and introducing disorder in specified portions: complete order (0D), less than one-quarter disorder (<1/4D), one-quarter disorder (1/4D), two adjacent quarters disorder (1/2AD), and two non-adjacent quarters disorder (1/2ND). The classifications and their corresponding disorder levels are illustrated in Figure 1. The five levels of disorder were carefully designed to systematically manipulate perceptual organization. All the elements in the image were controlled for luminance, size, shape, and color to minimize any unintended spatial attentional biases. Furthermore, the disordered regions across all image types were balanced to ensure an even distribution across different quadrants and orientations. For instance, in 1/4D images, the disordered region was balanced across the four quadrants, while in 1/2AD images, the disordered parts were balanced to appear alternately on the left, right, top, and bottom halves of different images. This balanced design ensured that any observed effects could not be attributed to a fixed spatial preference for the disordered portion of the image.

#### 2.1.3. Design and Procedure

Experiments 1a and 1b used a spatial cueing paradigm ([49]) and a 2 × 2 × 5 within-subjects design. This design included two levels of cue validity (valid/invalid), two image positions (Experiment 1a: left/right, Experiment 1b: top/bottom), and five disorder levels (0D, <1/4D, 1/4D, 1/2AD, and 1/2ND).

Figure 2 shows the experimental procedure for both Experiment 1a and Experiment 1b. In Experiment 1a, a fixation point (cross) was displayed at the center of the screen for 500-1000 ms. Next, a randomly selected image, either ordered or disordered and measuring 8.43° × 8.43°, was presented as the cue. The image appeared horizontally on the screen, which was positioned randomly to the left or right of the fixation point, with its center located 7.86° from the fixation point. Each image was displayed for either 100 ms or 200 ms, with both durations equally represented across 50% of the trials. After a 50 ms interval with a fixation point displayed, a probe dot (1.30° × 1.30°) appeared for 200 ms at either the same location as the image (valid cue) or the opposite location (invalid cue). The proportion of valid to invalid cues was balanced at 50% across all trials, ensuring that participants experienced an equal number of both cue types. This design enabled us to investigate the effects of cue validity on response times and accuracy. The design is used to study how spatial attention is allocated in response to visual cues, influenced not only by the spatial positions of the cues but also by the varying levels of disorder in the images. Additionally, this approach allowed for the examination of potential interaction effects between cue validity and disorder level on participants’ attentional responses.

Participants were instructed to press the “F” key for the left probe dot and the “J” key for the right probe dot but did not have to respond to the image (cue). Response times and accuracy were recorded, with each trial having a response deadline of 1500 ms. Experiment 1b followed the same procedure as Experiment 1a, except that the images were presented vertically, at either the top or bottom of the screen, which was positioned 7.86° away from the fixation point. Participants pressed the “T” key for a probe dot presented at the top of the screen and the “N” key for a probe dot presented at the bottom of the screen. In both Experiment 1a and Experiment 1b, each participant completed 320 trials and was instructed to respond as quickly and accurately as possible.

### 2.2. Results

The data were analyzed using IBM SPSS Statistics 26.0. Initially, mean reaction times (RTs, in milliseconds or ms) and mean error (in %) rates were calculated for each participant. Trials with RTs falling beyond ±2.5 *SD*s from the participant’s mean were excluded, which, on average, accounted for 3.6% of trials in Experiment 1a and 4.1% in Experiment 1b. Repeated measures ANOVA was conducted on the mean RTs and mean error rates. The within-subjects factors included cue validity (valid/invalid), image position (left/right for Experiment 1a; top/bottom for Experiment 1b), and disorder level (0D, <1/4D, 1/4D, 1/2AD, and 1/2ND).

Results for RTs in Experiment 1a: The interaction effect between image position and disorder level was significant. *F*(4, 16) = 9.200, *p* < 0.001, η_p_^2^ = 0.697. A post hoc simple effects analysis specifically examining the effect of image position (left vs. right) at all disorder level conditions (0D, <1/4D, 1/4D, 1/2AD, and 1/2ND) revealed significant differences for the 1/2AD (*p* = 0.009) and 1/2ND (*p* = 0.005) conditions. RTs were significantly slower for the Left position (1/2AD: *M* = 364.07 ms, *SE* = 18.94; 1/2ND: *M* = 364.00 ms, *SE* = 19.07) compared to the right position (1/2AD: *M* = 351.37 ms, *SE* = 17.54; 1/2ND: *M* = 349.00 ms, *SE* = 17.76). No significant differences were found for other disorder levels (all *ps* > 0.26). The interaction is depicted in Figure 3.

None of the other main effects or interactions reached significance (all *ps* > 0.12).

Results for error rates in Experiment 1a: The main effect of image position reached significance, *F*(1, 19) = 6.181, *p* = 0.022, η_p_^2^ = 0.245, with on average more errors when the image was displayed on the left of the screen (*M* = 4.09%, *SE* = 0.42%) compared to the right of the screen (*M* = 2.97%, *SE* = 0.42%). None of the other main effects or interactions reached significance (all *ps* > 0.15).

Results for RTs in Experiment 1b: A significant main effect of image position was observed, *F*(1, 19) = 7.171, *p* = 0.015, η_p_^2^ = 0.274, with RTs on average being faster when the image was displayed on the bottom of the screen (*M* = 370.893, *SE* = 13.427) compared to the top of the screen (*M* = 382.622, *SE* = 13.143). None of the other main effects or interactions reached significance (all *ps* > 0.06).

Results for error rates in Experiment 1b: No significant main effects or interactions were found, all *ps* > 0.12.

### 2.3. Discussion

In Experiments 1a and 1b, we used a spatial cueing paradigm to examine attentional biases and to implicitly activate associations between spatial positions and temporal concepts ([61]; [66]). In other words, using this paradigm could reduce any direct, goal-directed focus on the stimuli or the position, as the task required responses to the probe dot independently of stimulus presentation. Therefore, differences in reaction times and accuracy across stimulus positions can be attributed to participants’ ingrained spatial–temporal attentional biases. Experiment 1a revealed a higher error rate for probes following left-sided stimuli compared to right-sided stimuli. Analysis of the *discrepancy score of image position* further showed a rightward attentional bias under higher disorder levels (1/2AD and 1/2ND), which was consistent with the error rate results. This rightward bias aligns with the future-oriented spatial metaphor, as participants from left-to-right reading cultures tend to associate the right side with the “future” and the left side with the “past” ([41]; [47]; [54]). Interestingly, a leftward bias trend was observed in the 1/4D condition, which deviated from the expected pattern, as shown in Figure 3. This result suggested that intermediate disorder levels might impose unique visual or attentional demands, temporarily overriding typical spatial–temporal associations. However, as the underlying mechanisms remain unclear, further research is needed to explore how varying levels of visual complexity interact with attentional allocation and spatial–temporal associations. Experiment 1b showed faster responses to bottom-presented images compared to top-presented images, indicating a downward attentional bias. However, this effect was less pronounced than the horizontal orientation results, which may reflect a less stable mental timeline for vertical spatial representation. This finding is consistent with previous studies reporting varied vertical attentional biases ([5]; [15]; [28]; [58]). Additionally, the downward bias could be partially attributed to the visual system’s natural preference for the lower visual field, which is evolutionarily associated with detecting potential threats in the environment ([16]; [26]; [56]).

Moreover, the main effect of cue validity was not significant in either Experiment 1a or 1b, which might suggest a potential inhibition of return (IOR). The IOR is a cognitive process where attention is less likely to return to a previously attended location, leading to slower responses or more errors for validly cued positions ([48]). Typically, if attention were consistently maintained at the cued location, RTs for validly cued targets would be significantly faster than for invalidly cued ones, and error rates would be significantly lower. The absence of such significant differences in the present experiments suggested that attention may have been suppressed at previously attended locations, although the extent of this suppression appears to be mild. One potential explanation for this result is that the stimuli used in these experiments possess fewer complexity factors and adhere to an ordered structure reminiscent of Gestalt perception, facilitating rapid processing within short stimulus durations. An ERP study has demonstrated that Gestalt organization leads to swift processing occurring approximately 100 ms after stimulus onset ([27]), providing evidence for this explanation. Such rapid processing may reduce the time window during which attentional orienting effects, such as cue validity, can be robustly observed, thereby contributing to the lack of a significant cue validity effect.

## 3. Experiment 2

### 3.1. Method

#### 3.1.1. Participants

The minimal sample size was determined using the same method as in Experiments 1a and 1b. A power analysis was conducted to determine the required sample size for the study. It was estimated that a minimum of 24 participants would be necessary to achieve a statistical power of 80% for detecting medium effect sizes (*f* = 0.25) at a significance level of 5%. This sample size was specifically calculated to assess the 2 (image similarity) × 4 (image position) within-subject interaction. Thirty-five college students aged 18 to 29 years (5 males, *M*_age_ = 21.69 years, *SD* = 2.23) participated, 34 right-handed and one left-handed.

#### 3.1.2. Apparatus and Stimuli

The apparatus and stimuli were identical to those used in Experiments 1a and 1b.

#### 3.1.3. Design and Procedure

The design of Experiment 2 was a 2 × 2 × 4 within-subjects design, incorporating two levels of cue validity (valid/invalid), two levels of image similarity (same/different), and four image positions (left/right/top/bottom). Experiment 2 also used a spatial cueing paradigm, but unlike Experiments 1a and 1b, a central image was introduced before the cue image. This central image could be either identical to or different from the cue image, as shown in Figure 4.

Experiment 2 consisted of two parts with different orientations: horizontal (left/right) and vertical (top/bottom). Each participant first completed the horizontal part, then the vertical part, with 320 trials per part, totaling 640 trials per participant. Each trial began with a fixation point at the center of the screen, displayed for 500–1000 ms. Following the fixation, a central image appeared at the screen’s center for 500 ms. This central image was randomly chosen from five types (0D, <1/4D, 1/4D, 1/2AD, and 1/2ND, see Experiment 1). Next, a cue image was presented for either 100 ms or 200 ms. This cue image could appear on the left or right side (horizontal orientation) or on the top or bottom side (vertical orientation) of the screen. The cue image could be the same as (50% of the trials) or different from (50% of the trials) the central image but would always have the same disorder level as the central image. Thus, the two levels of image similarity (same/different) represent whether the cue image was completely identical to the central image or not. Trials were presented randomly. After a brief interval of 50 ms, in which the fixation cross was displayed again, a probe dot appeared for 200 ms. Participants responded to the probe’s location using the same keys as in Experiment 1a and 1b: “F” for left, “J” for right, “T” for top, and “N” for bottom. By structuring the experiment into two distinct parts and ensuring each participant completed both, the design aimed to comprehensively assess the effects of different image positions and cue validity on RTs and error rates. This approach allowed for a robust analysis of spatial attention across both horizontal and vertical orientations.

### 3.2. Results

The exclusion criteria for Experiment 2 were consistent with those used in Experiments 1a and 1b, where trials deviating more than ±2.5 *SD*s from the individual participant’s mean were excluded, accounting for an average of 4.0% of total trials. Repeated measures ANOVA was performed on the RTs and error rates, with cue validity (valid/invalid), image similarity (same/different), and image position (left/right/top/bottom) as within-subjects factors.

Results for RTs: The main effect of cue validity was significant, *F*(1, 34) = 59.136, *p* < 0.001, η_p_^2^ = 0.635. RTs were significantly slower for valid cue images (*M* = 363.99 ms, *SE* = 8.05) compared to invalid ones (*M* = 341.63 ms, *SE* = 8.83). This finding suggested the occurrence of an inhibition of return (IOR). The main effect of image similarity was significant, *F*(1, 34) = 10.086, *p* = 0.003, η_p_^2^ = 0.229, with RTs being faster when the cue image matched the central image (*M* = 351.90 ms, *SE* = 8.61) than when they differed (*M* = 353.72 ms, *SE* = 8.63). The main effect of image position was also significant, *F*(3, 32) = 13.028, *p* < 0.001, η_p_^2^ = 0.550. Post hoc comparisons using Bonferroni correction indicated that RTs were significantly faster when the image was presented on the left (*M* = 342.89 ms, *SE* = 9.555) or right (*M* = 336.50 ms, *SE* = 8.98) compared to the top (*M* = 367.83 ms, *SE* = 8.74) and bottom (*M* = 364.03 ms, *SE* = 8.73), *ps* < 0.001. There were no significant differences in RT between any other pair of image positions, *ps* > 0.16. Combining the data for the left and right positions into a horizontal orientation and the data for the top and bottom positions into a vertical orientation revealed a significant difference in RTs between these orientations, *t*(1, 34) = −6.360, *p* < 0.001, with RT being significantly faster in the horizontal orientation (*M* = 339.69 ms, *SE* = 9.17) compared to the vertical orientation (*M* = 365.93 ms, *SE* = 8.54).

A significant interaction effect was found between image similarity and image position, *F*(3, 32) = 3.617, *p* = 0.024, η_p_^2^ = 0.253. A post hoc simple effects analysis specifically examining the effect of image similarity (same vs. different) across image position levels (right, left, top, bottom) revealed significant differences in reaction times (RTs). When the central and cue images were the same, RTs were significantly faster at the right (*p* = 0.008) and top (*p* = 0.026) positions (Right: *M* = 334.94 ms, *SE* = 9.10; Top: *M* = 366.23 ms, *SE* = 8.60) compared to when the cue and central image were different (Right: *M* = 338.07 ms, *SE* = 8.89; Top: *M* = 369.44 ms, *SE* = 8.93). No significant differences were observed when comparing the same and different image conditions at Left and Bottom positions (all *ps* > 0.09). The interaction is also depicted in Figure 5.

None of the other interactions reached significance (all *ps* > 0.11).

Results for error rates: The main effect of cue validity was significant, *F*(1, 34) = 10.807, *p* = 0.002, η_p_^2^ = 0.241, with higher error rates when the cue image was valid (*M* = 4.51%, *SE* = 0.39%) compared to when it was invalid (*M* = 3.33%, *SE* = 0.24%), thus indicating an IOR. The main effect of the image position was also significant, *F*(3, 32) = 5.080, *p* = 0.005, η_p_^2^ = 0.323. Post hoc comparisons using Bonferroni correction indicated that error rates were significantly lower when the image was presented on the left (*M* = 3.54%, *SE* = 0.34%) than on the top (*M* = 4.70%, *SE* = 0.44%), *p* = 0.009; similarly, error rates were lower when the image was presented on the right (*M* = 3.30%, *SE* = 0.34%) than on the top (*p* = 0.014) or the bottom (*M* = 4.14%, *SE* = 0.31%, *p* = 0.047). There were no significant differences in error rates between any other pairs of image positions, *ps* > 0.29. Combining the data for the left and right positions into a horizontal orientation and the data for the top and bottom positions into a vertical orientation revealed a significant difference in error rates between these orientations, *t*(1, 34) = −2.768, *p* = 0.009, with error rates being significantly lower in the horizontal orientation (*M* = 3.73%, *SE* = 0.307%) compared to the vertical orientation (*M* = 4.42%, *SE* = 0.333%).

A significant interaction effect was also found between cue validity and image position, *F*(3, 32) = 3.414, *p* = 0.029, η_p_^2^ = 0.242. A post hoc simple effects analysis specifically examining the effect of cue validity (valid vs. invalid) across image position levels (left, right, top, bottom) revealed significant differences in error rates (Left: *p* < 0.001; Right: *p* = 0.004). At the left and right positions, error rates for the valid cue image (Left: *M* = 4.75%, *SE* = 0.60; Right: *M* = 4.07%, *SE* = 0.40) were significantly higher compared to the invalid cue image (Left: *M* = 2.32%, *SE* = 0.27; Right: *M* = 2.54%, *SE* = 0.27), which indicates an inhibition of return (IOR). No significant differences were found when comparing error rates for valid and invalid cue conditions at the top and bottom positions (all *ps* > 0.53). The interaction is depicted in Figure 6. Furthermore, the IOR data from the left and right positions were combined to represent the horizontal orientation, while the data from the top and bottom positions were combined for the vertical orientation. Analysis of these combined data revealed a significant difference, with the IOR being larger in the horizontal orientation (*M* = 1.98%, *SE* = 0.414%) compared to the vertical orientation (*M* = 0.38%, *SE* = 0.477%), *t*(1, 34) = 3.024, *p* = 0.005.

None of the other main effects or interactions reached significance (all *ps* > 0.12).

### 3.3. Discussion

Consistent with our expectations, participants responded faster when the central image matched the cue image, indicating that short-term memory enhances the processing of matching stimuli. Notably, the significant inhibition of return (IOR) was observed in Experiment 2, with participants responding more slowly and less accurately to valid cues. This suggested that memory representations enhanced attentional engagement as well as the amplified IOR, even though only half of the trials used the same central and cue images. In contrast, the lack of memory representations in Experiments 1a and 1b likely reduced attentional engagement and the IOR, resulting in no significant differences between valid and invalid cues. These findings align with previous research, which shows memory representations enhance attentional engagement ([7]; [20]; [60]); a further IOR inhibits the return of attention to previously attended locations.

Regarding the orientation, the result demonstrated faster responses and lower error rates in the horizontal orientation than in the vertical orientation. This finding suggested proficiency in processing horizontal information, likely due to common horizontal eye movements and attentional shifts in everyday activities such as reading and scanning environments ([31]; [74]). Moreover, the IOR was more pronounced in the horizontal orientation than in the vertical, which might be due to greater susceptibility to habituation in the horizontal direction ([59]).

A significant interaction between image similarity and image position was found in the results of reaction times (RTs). When the cue and central images were the same and positioned on the right and top, responses were faster. However, when the cue images were positioned on the left and bottom, there was no significant difference between the same and different conditions. These results indicate that the enhancement of short-term memory for matching stimuli varies by position, with a stronger facilitative effect observed in the right and top conditions.

## 4. Experiment 3

### 4.1. Method

#### 4.1.1. Participants

The minimal sample size was determined using the same method as in the previous experiments. A power analysis was conducted to determine the required sample size for the study. It was estimated that a minimum of 24 participants would be necessary to achieve a statistical power of 80% for detecting medium effect sizes (*f* = 0.25) at a significance level of 5%. This sample size was specifically calculated to assess the 2 (image similarity) × 4 (image position) within-subject interaction. Thirty-three college students aged 18 to 26 years (6 males, *M*_age_ = 21.22 years, *SD* = 2.61) participated, 31 right-handed and two left-handed.

#### 4.1.2. Apparatus, Design, Stimuli and Procedure

The apparatus in this experiment was identical to those used in Experiments 1 and 2. The design of Experiment 3 was similar to that of Experiment 2, featuring a 2 × 2 × 4 within-subjects design: 2 levels of cue validity (valid/invalid), 2 levels of image similarity (same/different), and 4 image positions (left/right/top/bottom). The stimuli and procedure for Experiment 3 mirrored that of Experiment 2, with the following key differences: when the cue image differed from the central image, it was not an image with the same disorder level as the central image but a completely ordered image. Additionally, the central image no longer included a completely ordered image but only partially disordered images at four different levels (<1/4D, <1/4D, 1/2AD, 1/2ND). Participants first completed the horizontal orientation of the experiment, followed by the vertical orientation, with 256 trials in each part, for a total of 512 trials per participant.

### 4.2. Results

The exclusion criteria for Experiment 3 were aligned with those of the previous experiments, excluding trials that accounted for an average of 4.6% of the total trials. Repeated measures ANOVA was conducted on the RTs and error rates, with cue validity (valid/invalid), image similarity (same/different), and image position (left/right/top/bottom) as within-subjects factors.

Results for RTs: Cue validity had a significant main effect, *F*(1, 32) = 75.495, *p* < 0.001, η_p_^2^ = 0.702. The RTs were significantly slower for valid cue images (*M* = 357.49 ms, *SE* = 7.03) compared to invalid ones (*M* = 334.40 ms, *SE* = 7.92), thus indicating an IOR. None of the other main effects or interactions reached significance (all *ps* > 0.08). Furthermore, combining the data for the left and right positions into a horizontal orientation and the data for the top and bottom positions into a vertical orientation revealed a significant difference in RTs between these orientations, *t*(1, 32) = −2.366, *p* = 0.024, with RTs being significantly faster in the horizontal orientation (*M* = 339.32 ms, *SE* = 8.74) compared to the vertical orientation (*M* = 352.57 ms, *SE* = 6.93).

Results for error rates: The main effect of cue validity was significant, *F*(1, 32) = 13.093, *p* = 0.001, η_p_^2^ = 0.290, with error rates significantly higher when the cue image was valid (*M* = 5.48%, *SE* = 0.44%) compared to when it was invalid (*M* = 3.89%, *SE* = 0.33%), thus indicating an IOR. A significant main effect of image position was also found, *F*(3, 30) = 6.652, *p* = 0.001, η_p_^2^ = 0.399. Post hoc comparisons using Bonferroni correction indicated that error rates were significantly smaller when the image was presented on the left (*M* = 4.21%, *SE* = 0.37%) compared to the top (*M* = 5.82%, *SE* = 0.43%), *p* = 0.002; and when presented on the right (*M* = 4.07%, *SE* = 0.43%) compared to the top, *p* = 0.004. Combining the data for the left and right positions into a horizontal orientation and the data for the top and bottom positions into a vertical orientation revealed a significant difference in error rates between these orientations, *t*(1, 32) = −3.922, *p* < 0.001, with error rates being significantly smaller in the horizontal orientation (*M* = 4.14%, *SE* = 0.340%) compared to the vertical orientation (*M* = 5.23%, *SE* = 0.352%).

A significant interaction effect between image similarity and image position was also observed, *F*(3, 30) = 3.672, *p* = 0.023, η_p_^2^ = 0.269. A post hoc simple effects analysis specifically examining the effect of image similarity (same vs. different) across image position levels (left, right, top, bottom) revealed significant differences in error rates. At the Left position, error rates were significantly larger when the central image and the cue image were the same (*M* = 4.97%, *SE* = 0.52) than in the different conditions (*M* = 3.46%, *SE* = 0.40), but only at the Left position, *p* = 0.008. No significant differences were found in other comparisons (all *ps* > 0.20). The interaction is also depicted in Figure 7.

None of the other interactions reached significance (all *ps* > 0.48).

### 4.3. Discussion

In line with the findings of Experiment 2, Experiment 3 demonstrated an IOR in both RTs and error rates. Responses were also faster and more accurate in the horizontal orientation compared to the vertical orientation. Moreover, the error rate results revealed a significant interaction effect between image similarity and image position. When images on the left matched the central image, error rates were higher compared to non-matching images. This pattern was different from the results for images on the right, top, and bottom, with a significant difference observed between the left and bottom positions.

This finding suggests that in the Left position, there is significant response inhibition for images represented in short-term memory compared to those that are not represented in short-term memory. Thus, our findings reveal a more nuanced interaction between image similarity and spatial attention. This contrasts with traditional priming effects, which typically involve broader cognitive processing. Traditional priming effects typically involve broader cognitive processing ([62]), and might involve a general facilitation of responses when a cue is presented, leading to faster reaction times regardless of spatial position or visual similarity. For example, in a word-priming task, presenting the word “doctor” might speed up the recognition of the word “nurse” due to their semantic association without any consideration of their spatial arrangement or visual features. However, our design allows us to go beyond this by specifically examining how the congruence of visual features (i.e., similarity) interacts with spatial positioning to influence cognitive processing. This approach reveals a more nuanced mechanism of attentional modulation, where the spatial location of stimuli and their visual similarity jointly determine the strength and direction of attentional effects. The more errors and increased inhibition when the cue and central image were the same on the left side suggest a unique mechanism of attentional modulation. It supports Liu et al.’s finding ([41]) from another perspective, suggesting that attention can be influenced not only by the temporal properties of the positions but also by the temporal information stored in the objects at those locations.

Interestingly, unlike Experiment 2, this experiment did not reveal a significant main effect of image similarity in either RTs or error rates. This result suggested that there was no significant difference in responses when the central and cue images were the same or different. This finding aligned with the possibility that both the short-term memory representations of the central image and fully ordered structures, which conform to Gestalt principles, may equally capture attention. As a result, when both are present simultaneously, any potential attention bias might be balanced out.

### 4.4. Combined Results of Experiment 2 and Experiment 3

As Experiment 2’s central image included five levels of disorder, whereas Experiment 3’s central image included only four levels of disorder (excluding the fully ordered images), we first excluded the trials where the central image was 0D for each participant in Experiment 2 to ensure that both experiments had the same experimental design and number of trials (512 trials). Repeated measures ANOVA was performed on the RTs and error rates, with cue validity (valid/invalid), image similarity (same/different), and image position (left/right/top/bottom) as within-subjects factors and the group (Experiment 2/Experiment 3) as a between-subjects factor.

Results for RTs: The main effect of cue validity was significant, *F*(1, 66) = 132.182, *p* < 0.001, η_p_^2^ = 0.667, with RTs significantly slower for valid cue images (*M* = 360.74 ms, *SE* = 5.61) compared to invalid ones (*M* = 338.02 ms, *SE* = 5.95), thus indicating an IOR. The main effect of image similarity was significant, *F*(1, 66) = 7.862, *p* = 0.007, η_p_^2^ = 0.106, with RTs being faster when the cue image matched the central image (*M* = 348.70 ms, *SE* = 5.69) than when they differed (*M* = 350.06 ms, *SE* = 5.72). A significant main effect of the image position was also found, *F*(3, 64) = 11.172, *p* < 0.001, η_p_^2^ = 0.344. Post hoc comparisons using Bonferroni correction indicated that RTs were significantly faster when the image was presented on the left (*M* = 342.45 ms, *SE* = 6.70) or right (*M* = 336.56 ms, *SE* = 6.19) compared to the top (*M* = 360.60 ms, *SE* = 5.71) or bottom (*M* = 357.91 ms, *SE* = 5.63), *ps* < 0.001. There were no significant differences in RTs between any other pair of image positions, *ps* > 0.08. Combining the data for the left and right positions into a horizontal orientation and the data for the top and bottom positions into a vertical orientation revealed a significant difference in RTs between these orientations, *t*(67) = −5.675, *p* < 0.001, with RTs being significantly faster in the horizontal orientation (*M* = 339.51 ms, *SE* = 6.30) compared to the vertical orientation (*M* = 359.45 ms, *SE* = 5.55).

A significant interaction effect was found between cue validity and image position, *F*(3, 64) = 4.699, *p* = 0.005, η_p_^2^ = 0.181. A post hoc simple effects analysis specifically examining the effect of cue validity (valid vs. invalid) across image position levels (left, right, top, bottom) revealed significant differences in reaction times (RTs). Across all spatial locations, RTs for the valid cue image (Left: *M* = 352.60 ms, *SE* = 6.54; Right: *M* = 348.89 ms, *SE* = 6.27; Top: *M* = 370.42 ms, *SE* = 5.42; Bottom: *M* = 371.07 ms, *SE* = 5.68) was significantly slower than that for the invalid cue image (Left: *M* = 332.30 ms, *SE* = 7.01; Right: *M* = 324.24 ms, *SE* = 6.35; Top: *M* = 350.78 ms, *SE* = 6.17; Bottom: *M* = 344.74 ms, *SE* = 5.91) across all positions (*ps* < 0.001), thus indicating an IOR. The interaction is also depicted in Figure 8.

None of the other interactions reached significance (all *ps* > 0.14).

Results for error rates: Cue validity had a significant main effect, *F*(1, 66) = 24.079, *p* < 0.001, η_p_^2^ = 0.267, with error rates significantly higher when the cue image was valid (*M* = 5.00%, *SE* = 0.29%) compared to when it was invalid (*M* = 3.61%, *SE* = 0.20%), thus indicating an IOR. A significant main effect of image position was also found, *F*(3, 64) = 11.785, *p* < 0.001, η_p_^2^ = 0.356. Post hoc comparisons using Bonferroni correction indicated that error rates were significantly smaller when the image was presented on the left (*M* = 3.88%, *SE* = 0.25%) compared to the top (*M* = 5.26%, *SE* = 0.31%), *p* < 0.001; and when presented on the right (*M* = 3.69%, *SE* = 0.24%) compared to both the top (*p* < 0.001) and the bottom (*M* = 4.39%, *SE* = 0.25%, *p* = 0.025).

A significant three-way interaction effect was found between image similarity, image position, and Group, *F*(3, 64) = 3.915, *p* = 0.012, η_p_^2^ = 0.155. A post hoc simple effects analysis was conducted to examine the error rates across different experimental conditions, specifically comparing image similarity (same vs. different) within each image position level (left, right, top, bottom) between Experiment 2 and Experiment 3. When the central image and the cue image were the same at the Left position, the error rates were higher in Experiment 3 (*M* = 4.97%, *SE* = 0.46%) than in Experiment 2 (*M* = 3.39%, *SE* = 0.45%), *p* = 0.016. No significant differences were found in other comparisons (all *ps* > 0.15). The interaction is also depicted in Figure 9.

None of the other main effects or interactions reached significance (all *ps* > 0.17).

### 4.5. Discussion

The results combining Experiments 2 and 3 were consistent with those of the two separate experiments. Firstly, the IOR was found in both RTs and error rates. Furthermore, the RT results indicated that participants responded faster to horizontal orientations than to vertical orientations and faster when the cue image matched the central image compared to when they were different. Additionally, a significant group difference was found in both RTs and error rates, with Experiment 2 showing slower response and higher accuracy than Experiment 3. The faster response in Experiment 3 might be due to the use of fully ordered images as the different cue images, leading to faster information processing. The accompanying lower accuracy could be due to the faster responses (i.e., speed–accuracy tradeoff ([12]).

Notably, the analysis of error rates revealed a three-way interaction effect among image similarity, image position, and the group. Specifically, in Experiment 3, the error rate was higher when the cue image was presented on the left and was the same as the central image, compared to Experiment 2 (see Figure 9). This difference may be attributed to the fact that, in Experiment 3, the cue images different from the central images were complete Gestalt structures. This finding suggests that when cue images are processed more efficiently (e.g., fully ordered images that are easier to process and require less cognitive effort ([43])), structures represented in short-term memory are more strongly suppressed at locations metaphorically associated with past temporal meanings. As shown in Figure 9, error rates under the same conditions for the left and top positions in Experiment 3 were consistently higher than those in Experiment 2, with the increase on the left side being particularly pronounced.

## 5. General Discussion

The present study reveals how the *spatial metaphor of time* shapes attentional bias, as well as how visual short-term memory representations and Gestalt perceptual grouping modulate this effect. Our investigation focused on three main questions: how attentional bias shifts with spatial orientation associated with the temporal concept, how short-term memory representations influence this bias, and whether short-term memory representations affect attention similarly to fully ordered structures aligned with Gestalt principles. These findings shed new light on the dynamic roles of spatial cues and memory in guiding attention.

The first finding supported the *spatial metaphor of time* ([6]; [47]; [54]; [65]), suggesting a possible inherent future-oriented bias in attentional processing. Specifically, we observed a marked rightward attentional bias in the horizontal orientation in Experiment 1a. This rightward bias aligns with the common cultural representation of time as moving from left to right ([47]; [54]), suggesting that our cognitive processes might be influenced by these spatial–temporal metaphors. The spatial cueing paradigm employed in this study effectively activated implicit spatial–temporal metaphors by relying solely on participants’ responses to probe dots at different positions without inducing additional cognitive operations related to stimuli or spatial factors. This approach allowed us to detect implicit spatial–temporal associations through differences in reaction times and error rates. While previous research has focused on memory-driven, past-oriented attention ([2]; [18]; [37]; [45]), our findings using the spatial cueing paradigm provided evidence that could suggest a natural future-oriented spatial bias. This is further supported by the fact that the rightward bias is consistent with the cultural notion of time progression ([47]; [54]). In Experiment 1b, the downward bias was less pronounced but, together with the findings from Experiment 1a, provided further evidence that spatial–temporal associations play a more dominant role in shaping attentional biases. This preference for future-related positions ([15]; [29]) highlights the consistent influence of spatial–temporal metaphors across different orientations. Additionally, the findings from Experiments 1a and 1b, together with recent work by [41] ([41]), could deepen our understanding of the *spatial metaphor of time*. These findings suggest that these metaphors might not merely represent symbolic mappings but could reflect functionally embedded biases within our perceptual systems. In other words, the future-oriented bias observed here may not just be an abstract concept (such as “left for past, right for future”) but an automatic attentional shift deeply rooted in perceptual processing. This directional bias might help us respond more quickly and effectively to future-related information, reflecting an adaptive mechanism that facilitates anticipatory processing and enhances our interactions with the environment.

When introducing short-term memory representations in Experiment 2, we found that participants responded faster and more accurately when the central image matched the cue image. This finding aligned with our expectation, suggesting that visual short-term memory elicited attentional bias and is consistent with previous research on the influence of working and sensory memory on attention ([7]; [20]; [60]; [73]). Both RTs and error rates showed a significant inhibition of return (IOR), which was absent in Experiment 1. This result suggested that short-term memory representations may play a key role in suppressing previously processed information and triggering an IOR. Although [52] ([52]) observed that attentional bias through object-based cueing can improve processing speed and accuracy by enhancing target detection and identification, our findings demonstrate that the IOR may counteract this benefit by limiting reorienting, especially when short-term memory cues are involved. Additionally, the stronger IOR observed in horizontal orientations, compared to vertical ones, likely reflects the impact of habitual horizontal task processing, which has been shown to enhance processing efficiency and attentional stability along this orientation ([44]; [50]).

Experiment 3 reinforced the findings from Experiment 2, showing similar IOR patterns in both RTs and error rates, as well as faster and more accurate responses in horizontal orientations compared to vertical ones. Furthermore, the results of Experiment 3 indicated that the suppression of short-term memory representations varies in magnitude across different locations. When the central and cue images were identical and the cue image was presented on the left, there was increased suppression at the cue image location. This finding provided insight into how short-term memory representations may selectively facilitate attentional engagement, intensifying suppression where these cues were presented. However, unlike in Experiment 2, there were no significant differences in RTs or error rates between the two levels of image similarity (where cue and central images were either identical or different). This contrast likely arose from the variations in the “different cue image” across experiments: in Experiment 3, the different cue image was a fully ordered structure, while in Experiment 2, it shared a similar level of disorder with the central image. The absence of a significant finding in Experiment 3 suggested that fully ordered and partially disordered structures held in short-term memory capture attention at a similar level. This outcome underscored the stability of fully ordered Gestalt structures in capturing attention, suggesting that their inherent perceptual organization can draw attentional resources comparably to short-term memory representations. As shown in studies on perceptual grouping, this process is largely pre-attentive, occurring even without spatial attention ([38]; [53]). Specifically, the Gestalt principle of proximity allows for cohesive groupings to form automatically, enhancing the perceptual stability of ordered stimuli. Moreover, a recent study found that neural representations of grouping progress across early visual areas, with V3 exhibiting stronger representational strength for grouping than V1 and V2. This grouping effect was further amplified when attention was directed toward the grouped stimuli, thus indicating that attentional engagement can provide top–down modulation in V3, boosting the effects of grouping ([72]).

The combined analysis of Experiments 2 and 3 confirmed not only a significant IOR but also the faster RTs when participants responded to cue images that matched the central image. This result suggested that representations maintained in short-term memory are processed more efficiently than other types of structures (including both completely ordered and partially disordered structures). In addition, there were noticeable differences in RTs and error rates between the groups. Responses in Experiment 3 were faster than in Experiment 2, yet the error rates were also higher. This discrepancy could be attributed to the speed–accuracy tradeoff, where faster processing of cue images in Experiment 3 may have accelerated responses to the probe dot but reduced accuracy ([12]; [30]; [68]).

Notably, both Experiments 2 and 3 observed unique characteristics when cue images were presented on the left compared to other positions. In Experiment 2, when the cue image was on the left and matched the central image, reaction times (RTs) tended to be longer than those under differing conditions. This trend contrasted with observations from the other three positions, particularly on the right and top. Additionally, the error rate results indicated a significant inhibition of return (IOR) effect on the left side. In Experiment 3, error rates were significantly higher for images on the left that matched the central image compared to those that did not match. This finding differed from the patterns observed for images on the right, top, and bottom. Additionally, a significant three-way interaction effect among image similarity, image position, and the group was found. In Experiment 2, the error rate was higher for cue images different from the central image when presented on the left, while in Experiment 3, the opposite was true. Moreover, when the cue image was on the left and matched the central image, the error rate was significantly higher in Experiment 3 compared to Experiment 2. These findings suggested a more pronounced IOR and suppression of structures represented in short-term memory when cue images are displayed on the left, particularly in Experiment 3, when the cue image differed from the central image and was completely ordered. This phenomenon is likely related to the psychological timeline, as demonstrated in Experiment 1, where individuals preferred future-related right space positions. Furthermore, the representation in short-term memory associated with the past and present on the left side might lead to compounded suppression effects, resulting in slower response or higher error rates. The differences in error rates on the left side between Experiments 2 and 3 further suggest that when the overall efficiency of processing cue images is high (i.e., due to the presentation of fully ordered images, which are easier to process and require less cognitive effort), there is stronger suppression of structures represented in short-term memory. This finding may be influenced by the predominance of right-handed participants in our study. Previous research has shown that right-handed individuals often perform significantly worse than chance on directional recall tasks, whereas left-handed individuals do not exhibit the same level of impairment ([33]). Given that recall tasks involve retrieving past-related information, this may point to a potential suppression of past-related information among right-handed individuals. Therefore, it is essential to explore how handedness affects cognitive processes related to temporal information. Future research could explore whether different temporal contexts (e.g., past versus future) elicit distinct effects based on handedness, offering valuable insights into how individuals process information.

Taken together, this study highlights the dynamic interplay between spatial–temporal metaphors, short-term memory representations, and Gestalt principles in shaping attentional biases. The findings provide compelling evidence for a future-oriented attentional bias, demonstrate the role of short-term memory in modulating attention, and reveal the influence of Gestalt perceptual grouping on attentional allocation. However, several limitations of the study warrant consideration. The stimuli employed in our experiments consisted of simple geometric shapes devoid of complex attributes or deeper semantic associations. Additionally, the experimental paradigm required participants to engage in relatively straightforward tasks. These design choices, while ensuring experimental control, may have constrained the ability to fully capture the potential effects of disorder on attention. Although Experiment 1a revealed that varying levels of disorder exert distinct influences on attentional biases, other underlying phenomena may not have been adequately captured or fully explained. Moreover, the generalizability of these findings remains limited and warrants further investigation. Future research should address these limitations by employing more complex and meaningful stimuli and refining experimental paradigms to better capture the nuanced effects of order/disorder on cognitive mechanisms such as attention and memory. Such efforts would not only advance our understanding of how the brain interacts with the environment but also offer significant practical applications. For instance, insights from this research could inform the design of user interfaces in high-stakes domains, such as air traffic control or autonomous vehicles, by aligning designs with human attentional preferences. Additionally, understanding the neural mechanisms underlying attentional allocation could lead to the development of interventions or training programs aimed at optimizing cognitive performance in demanding environments, ultimately improving safety and efficiency in real-world applications.

## 6. Conclusions

The present study reveals how spatial–temporal metaphors shape attentional biases and how visual short-term memory representations and Gestalt perceptual grouping modulate this effect. Specifically, it demonstrated a future-oriented preference across spatial orientations, with a distinct rightward bias in the horizontal orientation and a downward tendency in the vertical orientation. Short-term memory was shown to not only elicit attentional bias but also suppress previously attended information, which resulted in a significant inhibition of return (IOR). Furthermore, Gestalt principles were found to play a fundamental role in visual attention, which balances the attentional capture driven by memory representations. These findings deepen our understanding of how spatial–temporal cues, memory, and perceptual organization interact to influence attention and provide practical insights for optimizing interface design and decision-making systems in safety-critical contexts.

## Figures and Tables

**Figure 1 behavsci-15-00599-f001:**

Examples of stimuli used in Experiment 1. From left to right, the images represent five disorder levels: complete order (0D), less than one-quarter disorder (<1/4D), one-quarter disorder (1/4D), two adjacent quarters disorder (1/2AD), and two non-adjacent quarters disorder (1/2ND).

**Figure 2 behavsci-15-00599-f002:**
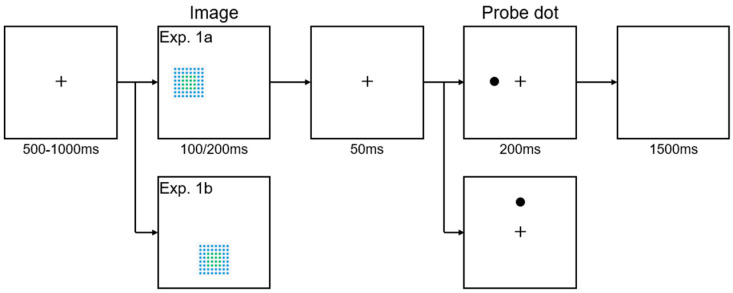
The procedure of Experiments 1a and 1b. In both experiments, participants responded to the position of probe dots. In Experiment 1a, they pressed “F” for dots on the left side of the screen (as shown in the image) and “J” for dots on the right. In Experiment 1b, they pressed “T” for dots at the top of the screen (as shown in the image), and “N” for dots at the bottom. The procedure for Experiment 1a shows a valid cue trial; the procedure for Experiment 1b shows an invalid cue trial.

**Figure 3 behavsci-15-00599-f003:**
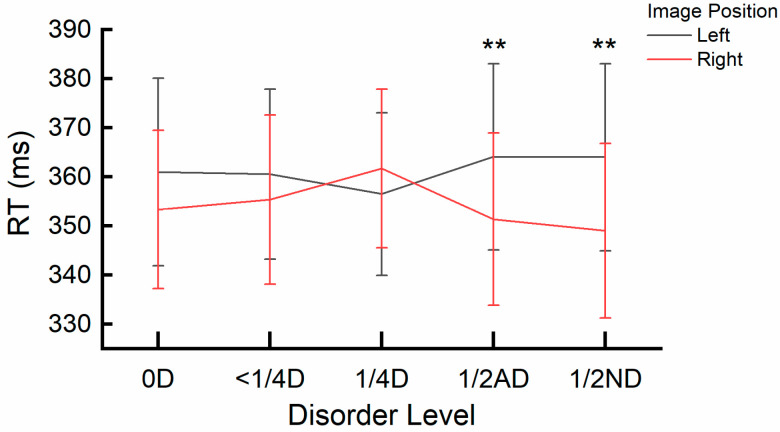
Interaction effect of image position and disorder level on reaction times (RTs) in Experiment 1a. ** (*p* < 0.01) reflects significant differences in the *discrepancy score of image position*. Error bars represent the Standard Error of the Mean (SEM).

**Figure 4 behavsci-15-00599-f004:**
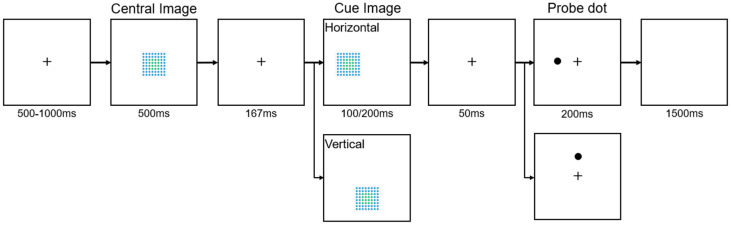
The procedure of Experiment 2. In this experiment, participants responded to the position of probe dots. In Experiment 1a, they pressed “F” for dots on the left side of the screen (as shown in the image) and “J” for dots on the right. In Experiment 1b, they pressed “T” for dots at the top of the screen (as shown in the image) and “N” for dots at the bottom. The procedure for the horizontal orientation shows a valid cue trial; the procedure for the vertical orientation shows an invalid cue trial.

**Figure 5 behavsci-15-00599-f005:**
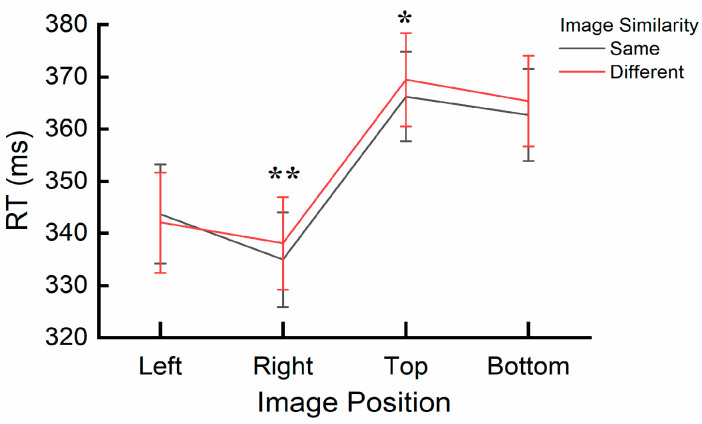
Interaction effect of image similarity and image position on reaction times (RTs) in Experiment 2. * (*p* < 0.05) and ** (*p* < 0.01) reflect the significant difference in the *discrepancy score of image similarity*. Error bars represent SEM.

**Figure 6 behavsci-15-00599-f006:**
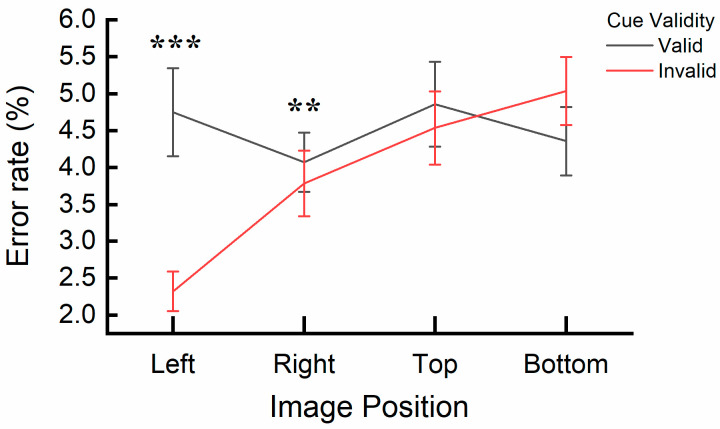
Interaction effect of cue validity and image position on error rates in Experiment 2. ** (*p* < 0.01) and *** (*p* < 0.001) reflect the significant difference related to the inhibition of return (IOR). Error bars represent SEM.

**Figure 7 behavsci-15-00599-f007:**
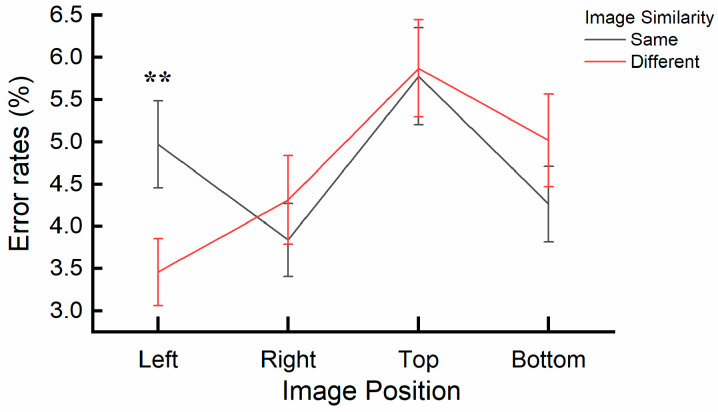
Interaction effect of image similarity and image position on error rates in Experiment 3. ** (*p* < 0.01) reflects the significant difference in the *discrepancy score of image similarity*. Error bars represent SEM.

**Figure 8 behavsci-15-00599-f008:**
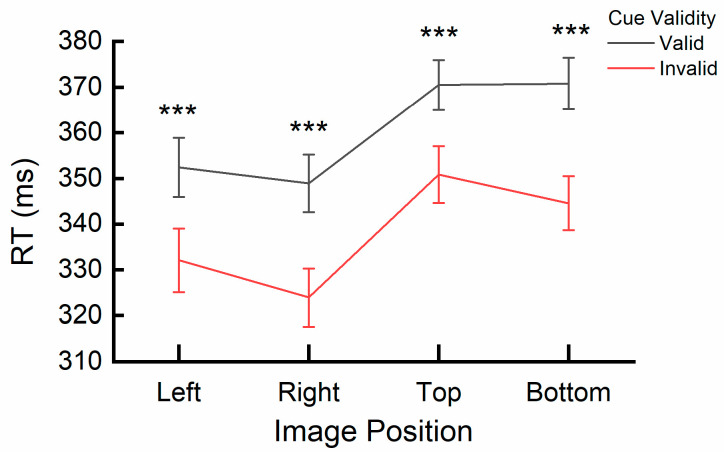
Interaction effect of cue validity and image position on error rates. *** (*p* < 0.001) reflects the significant difference related to the inhibition of return (IOR). Error bars represent SEM.

**Figure 9 behavsci-15-00599-f009:**
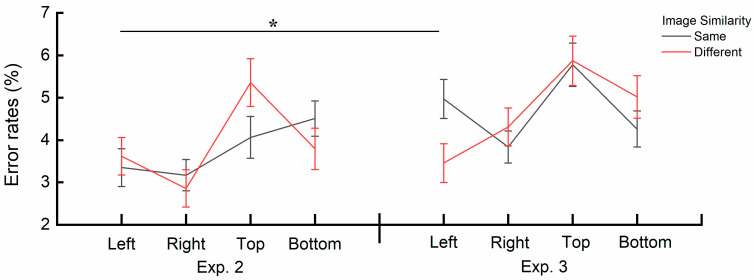
Three-way interaction effect among image similarity, image position, and group on reaction times (RTs). * (*p* < 0.05) reflects significant differences in the *discrepancy score of image similarity*. Error bars represent SEM.

## Data Availability

The stimuli and the experimental data, including both raw and cleaned data, are available on OSF (doi: 10.17605/OSF.IO/BDZ5K).

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
