# Peer review of "Attention Dynamics in Spatial–Temporal Contexts"

_behavsci, 2025, doi:10.3390/bs15050599_

Round 1
Reviewer 1 Report (Previous Reviewer 1)
Comments and Suggestions for Authors
I appreciate the changes made by the authors to the manuscript; however, I believe certain aspects still require modification or, at the very least, further clarification. Below is a list of suggested changes:
- Introduction: Line 81 – It would be better to clearly define the acronyms (1/2 AD and 1/2 ND) as this is the first instance they are presented and they have not yet been explained.
- Line 99: To facilitate reading, I would suggest explicitly stating the hypothesis underlying this expected result.
- Lines 154–155: It is unclear whether participants were required to respond to valid or invalid cues. The discussion section mentions that they were instructed to respond solely to the dot, regardless of the cue, making it unclear what the rationale was for using this type of Posner spatial cueing task if the cue itself was not taken into account.
- Figure 2 Caption: "for dots at the bottom (as shown in the image)" — no dots are present at the bottom in the image.
- Section 2.2 Results: Line 179 – The ANOVA was performed on the RTs, but were the post hoc tests conducted on the discrepancy scores? The same question arises in Section 3.2 Results (line 321).
- Line 195: (all ps > 0.013) — typo.
- Figure 3 Caption: It should be emphasized that the significant differences pertain to the discrepancy score; otherwise, the graph is unclear. This also applies to other images from the subsequent experiments — please check that the figure captions are coherent and clear.
- Section 2.3 Discussion: Was the rationale behind using a Posner task with a 50% probability for valid/invalid cues to mitigate the preference effect for stimuli presented on the right, through invalid cues on the left, given that participants were then only required to respond to the dot? If so, this rationale should be made explicit.
- Section 3.2 Results: Line 334 – (all ps > 0.012) — typo.
- Section 2.4 Discussion: Line 370 – A clearer explanation is needed for the difference between this effect and the priming effect.
- General Discussion: Line 575 – It remains unclear how the authors can assert that it is the spatial representation of time that induces the observed effect.
- Line 583: Since no words were used in this experiment, how does verbal processing fit into this study?
- Line 598: While I fully agree with the results concerning the spatial representation of time and the greater response facilitation toward the future (right) in the young participants tested, I find that there is still a logical gap. Why is the association framed as past/future when the introduction refers to "before/after"?
Author Response
Dear Reviewer,
Thank you for your thorough and constructive feedback on our manuscript. We greatly appreciate your acknowledgment of the changes we have implemented and your insightful suggestions for further enhancement. Your comments regarding the clarity of acronyms, the explicit articulation of our hypothesis, and the rationale behind our experimental design are especially valuable to us. We are committed to addressing these points in our revisions.
For a detailed account of the modifications made, please refer to the revised manuscript and our point-by-point response.
Thank you once again for your guidance.
Best regards,
Yuying Wang

Reviewer 2 Report (New Reviewer)
Comments and Suggestions for Authors
Summary
The manuscript “Attention Dynamics in Spatial-Temporal Contexts” examines how spatial metaphors for understanding time influence attentional biases for visual stimuli for (dis)ordered stimuli. Across three experiments, the authors investigate this question with manipulations related to degree of disorder (Experiments 1a and 1b), degree of memory involvement (Experiments 2 and 3), and interactions of (dis)order with memory (Experiment 3). The results overall showed a consistent preference for future-related information, particularly in the right visual field and horizontally oriented.
Evaluation
The experiments appear to be well done, and the results are interesting. I have one larger comment related to setting up the experiments more robustly in the Introduction, which are followed by several specific comments.
- Greater theoretical links to present experiments in the Introduction
The introduction is written clearly, but I had some trouble following the linking hypotheses provided by the authors. For example, the authors describe the construct of attention, including object-based attention in visual search paradigms. They then state that beyond understanding the spatial location of objects in visual search paradigms, it is crucial to understand the influence of the spatial metaphor of time. Could the authors provide an example of how this might be investigated to help make their point? For example, do the authors mean that something thought to be older would be more easily found in the left visual field of a visual search array?
As a related point, “Gestalt perceptual grouping” is quite a broad term that is not sufficiently defined in the Introduction. I recommend that the authors devote more space to setting up this construct, as it is an important component of the paper. I think this would help establish why the authors are manipulating disorder and how it relates in principle to attention and memory.
I hope that addressing these points will help the reader follow the logic / rationale of the study. At present, once I got to the specific hypotheses of the study (final paragraph of the Introduction), I was surprised by the prediction that participants were expected to exhibit an attentional bias toward stimuli on the right in the horizontal orientation. What, specifically about the spatial metaphor of time would lead to that prediction? Whatever it is, it should be established earlier in the Introduction.
Specific comments:
L11-12: The opening sentence of the abstract is quite hard to unpack. Consider breaking into two sentences (e.g., first sentence describing the purpose of the study, second sentence previewing the takeaways). Alternatively, I think the abstract reads quite well if the authors remove the final clause of the sentence (beginning with “highlighting”)
L81: I am not sure what the levels of disorder refer to (e.g., 0D, 1/2AD, 1/2ND) – please describe these in the prose (e.g., what is D versus AD versus ND) prior to their use here. I see that these are described subsequently (L122-125) but if the authors wish to use these shorthand descriptions, they need to be defined here.
L103: Please provide some more information here about the power analysis – specifically, which “F test” did you select in G*Power?
L194: I was struck by the sentence “No significant differences were observed in the discrepancy score of Image Position between any other pairs of disorder levels (all ps > .013).” Does this imply that some p-values were between .013 and .05? Are the authors referring to uncorrected p-values or Bonferroni corrected p-values? Overall, given the power analyses, which use a Type 1 error of .05, I was a little surprised to see a value in the parentheses that was below this threshold.
Author Response
Dear Reviewer,
Thank you for your thoughtful evaluation and constructive feedback on our manuscript. We sincerely appreciate your positive remarks regarding the quality of our experiments and the significance of our findings.
In response to your suggestion for stronger theoretical connections in the Introduction, we provided a more detailed explanation of how the principles of Gestalt perceptual grouping relate to attention and memory, thereby enhancing the overall coherence of the arguments presented. We have also revised the opening of the abstract, added explanations regarding the disorder levels in the Introduction, and included essential information about the determination of sample size in the Methods section. Furthermore, we clarified the reasoning behind considering a p-value of less than 0.05 as non-significant with Bonferroni correction.
Additionally, we have implemented detailed modifications throughout the manuscript to improve its overall quality and address your concerns.
For a comprehensive overview of the changes we have made, please refer to the revised manuscript along with our detailed point-by-point response.
Thank you once again for your valuable insights.
Best regards,
Yuying Wang

Round 2
Reviewer 1 Report (Previous Reviewer 1)
Comments and Suggestions for Authors
I greatly appreciate the work done by the authors and thank them for addressing my concerns. I believe that the paper has significantly improved in the clarity and the readability. However, I remain concerned about the methodology used for the statistical analyses. Specifically, I do not find it appropriate to conduct an ANOVA on reaction times and then perform post-hoc tests on a different derived measure, such as the discrepancy score. This approach introduces a methodological inconsistency that may compromise the interpretability of the results. ANOVA is designed to analyze variance across groups concerning a single dependent variable, and for post-hoc tests to be valid, they must be conducted on the same variable examined in the main analysis (Maxwell & Delaney, 2004). When post-hoc tests are performed on a different variable, they no longer reflect the same data distribution analyzed in the ANOVA, potentially leading to misleading or statistically invalid conclusions. Maintaining consistency between the main analysis and the post-hoc tests ensures that multiple comparisons are derived from the same statistical model, preventing interpretative biases.
Maxwell, S. E., & Delaney, H. D. (2004). Designing Experiments and Analyzing Data: A Model Comparison Perspective. Psychology Press.
Author Response
Dear Reviewer,
We are grateful for your comprehensive review and valuable suggestions. Your recognition of our manuscript's enhanced clarity is deeply appreciated. Addressing your methodological concerns, we have added a post-hoc simple effects analysis and revised the discussion section accordingly. These changes provide a more robust examination of the interaction effect and improves the overall statistical validity of our research. We hope these modifications address any remaining concerns. More details are available in our resubmission manuscript and point-by-point response letter.
Best regards,
Yuying Wang

Round 3
Reviewer 1 Report (Previous Reviewer 1)
Comments and Suggestions for Authors
I thank the authors for once again addressing my concerns. Regarding the discrepancy score as a measure, I fully agree with the possibility of using it, especially if it helps to explain the data. My concern, however, is related to the accuracy of the methodology and the consistency between the two analyses. If ANOVA and post hoc tests are reported for RTs, then ANOVA and post hoc tests for the discrepancy score should also be included.
Author Response
Dear reviewer,
I appreciate your thorough review and insightful feedback, which highlight the importance of methodological accuracy. Our further analysis revealed that the interaction effect on the discrepancy score was non-significant, prompting us to concentrate exclusively on the simple effects analysis. We are committed to maintaining a more rigorous approach in our data analysis for future studies.
Best regards,
Yuying Wang

Round 4
Reviewer 1 Report (Previous Reviewer 1)
Comments and Suggestions for Authors
I am pleased to read the revised version of the manuscript, which has been corrected from a methodological standpoint. The authors have addressed all of my comments satisfactorily.
This manuscript is a resubmission of an earlier submission. The following is a list of the peer review reports and author responses from that submission.
Round 1
Reviewer 1 Report
Comments and Suggestions for Authors
In the present study, the authors aimed to investigate the effects of attentional biases induced by the spatial metaphor of time, or spatial representation of time, as well as the impact of object representations maintained in short-term memory. In addition, the study examines how attention might be differentially affected by images perceived as unified entities, according to Gestalt principles, compared to images not organized following Gestalt principles when held in short-term memory.
The manuscript is generally well-written, and the English is clear and understandable. The statistical analyses are appropriately conducted. However, some sections require refinement to enhance the clarity of the study's objectives and hypotheses, and to better situate the work within its relevant segment of the literature.
Abstract: In this section, the authors report conducting six experiments, but subsequently refer to three experiments. It would be better to clarify this inconsistency upfront or maintain coherence by referring consistently to three experiments, as experiments 2 and 3 do not involve two separate sub-experiments but rather two tasks administered to the same participants. Furthermore, the reference to the varying "disorder levels" of the images used in the tasks is not explained here, leading to confusion. This is only clarified in the Materials and Methods section, but there are other references to this term in the Introduction, which hinders clarity.
Introduction: The introduction would benefit from reorganization to avoid redundant explanations. For example, the content in lines 69–71 is already covered earlier, and the explanation in lines 80–86 could be moved to lines 62–63 for better logical flow and clearer connection to subsequent hypotheses.
The logical progression between the importance of studying the effects of spatial representations of time on object-based attention and the stated objectives of the study is unclear. This connection is revisited later with the study goals but remains ambiguous. Additionally, there is no clear explanation of IOR in terms useful for understanding the subsequent results, making it difficult to follow the authors’ discussion.
The hypotheses are not clearly articulated in terms of computational models and cognitive processes, nor are they clearly linked to the experimental protocol described above. Specifically, hypothesis 2 could potentially be explained as a priming effect, which raises questions about how the authors' innovative hypothesis differs from this explanation.
Materials and Methods:
-
Experiment 1: Were all participants right-handed? Moreover, do the authors believe that the consistent gender imbalance across groups and experiments might have introduced any differences? Additionally, the main text does not state whether participants signed informed consent.
It is unclear whether the disorder in the conditions (particularly 1/4D and 1/2AD) always pertained to the right-hand side of the image (as shown in Fig. 2) or also to the left-hand side. Clarifying this would help disambiguate whether the effect is due to greater attention directed to the disordered quarter of the image. Additionally, it is unclear whether the intent of the experiment was to induce a Posner effect using the images in different positions. If so, the congruence and incongruence percentages of the cue relative to the actual presentation should be specified. Furthermore, the rationale behind the selection of these specific images, which could themselves induce spatial attentional biases, is not explained.
Regarding the results, the significance of the discrepancy score used to demonstrate the rightward effect of future events is not adequately explained. Merely defining how the score is calculated does not help integrate the information and understand the results. This applies throughout the manuscript whenever the term “discrepancy” or “difference” is used to describe results. If the significant differences reported in the figures pertain to the discrepancy score, it would be better to explicitly state this for easier interpretation.
In the discussion of Experiment 1, it is unclear how the rightward attentional bias is attributed to the future-oriented spatial representation of time and not to some other task-related effect. It is also unclear why this conclusion is drawn specifically from the observation that the image with disorder in the 1/4D condition induces an association between the right position and the future. The connection between a Posner paradigm and the spatial representation of time, which remains implicit in the task, is not made clear.
-
Experiment 2: What is the difference between the effect of image similarity and the priming effect? Additionally, the explanation of the IOR observed in the study is not well contextualized within the experiment.
-
Experiment 3: Line 434 contains a typo: it refers to error rates but seems to discuss reaction times (RT).
General Discussion: Lines 497–499: Based on the study’s results, how can the facilitation in responding to images presented on the right be conclusively attributed to their association with the future? While the theoretical foundation of the spatial representation of time is compelling, it is unclear whether the task employed truly supports this hypothesis. Moreover, the discussion would benefit from being expanded to include additional literature supporting the results provided by the study.
Author Response
Response to Reviewer 1 Comments
|
||
1. Summary |
|
|
In the present study, the authors aimed to investigate the effects of attentional biases induced by the spatial metaphor of time, or spatial representation of time, as well as the impact of object representations maintained in short-term memory. In addition, the study examines how attention might be differentially affected by images perceived as unified entities, according to Gestalt principles, compared to images not organized following Gestalt principles when held in short-term memory. The manuscript is generally well-written, and the English is clear and understandable. The statistical analyses are appropriately conducted. However, some sections require refinement to enhance the clarity of the study's objectives and hypotheses, and to better situate the work within its relevant segment of the literature. Response: Thank you very much for your thorough and insightful review of our manuscript. We greatly appreciate the time and effort you have invested in providing us with such detailed feedback. We have carefully addressed your comments and made revisions to improve the clarity and robustness of our study. |
||
2. Questions for General Evaluation |
Reviewer’s Evaluation |
Response and Revisions |
Is the content succinctly described and contextualized with respect to previous and present theoretical background and empirical research (if applicable) on the topic? |
Must be improved |
We sincerely appreciate your valuable feedback. In response to your comments, we have revised the introduction, methods, and discussion sections to more clearly articulate the theoretical background and experimental hypotheses. The research methods have been further detailed to ensure transparency and clarity. Additionally, we have refined the discussion to improve coherence and balance, ensuring that our findings are thoroughly contextualized within the results and relevant literature. These revisions aim to enhance the overall clarity, rigor, and impact of the manuscript.
|
Are the research design, questions, hypotheses and methods clearly stated? |
Must be improved |
|
Are the arguments and discussion of findings coherent, balanced and compelling? |
Must be improved |
|
For empirical research, are the results clearly presented? |
Can be improved |
|
Is the article adequately referenced? |
Must be improved |
|
Are the conclusions thoroughly supported by the results presented in the article or referenced in secondary literature? |
Must be improved |
|
3. Point-by-point response to Comments and Suggestions for Authors |
||
Comments 1: Abstract: In this section, the authors report conducting six experiments, but subsequently refer to three experiments. It would be better to clarify this inconsistency upfront or maintain coherence by referring consistently to three experiments, as experiments 2 and 3 do not involve two separate sub-experiments but rather two tasks administered to the same participants. Furthermore, the reference to the varying "disorder levels" of the images used in the tasks is not explained here, leading to confusion. This is only clarified in the Materials and Methods section, but there are other references to this term in the Introduction, which hinders clarity. |
||
Response 1: Thank you for your valuable feedback. We have revised the abstract to accurately reflect the experimental design, correcting the previous ambiguity regarding the number of experiments to clearly indicate the three distinct experiments undertaken. Regarding the reference to the varying "disorder levels" of the images used in the tasks, we have added a brief explanation in the abstract (page 1, line 12) to provide contextual clarity within the word limit. A more detailed explanation has been included in the Introduction (page 2, lines 81-84). |
||
Comments 2: Introduction: The introduction would benefit from reorganization to avoid redundant explanations. For example, the content in lines 69–71 is already covered earlier, and the explanation in lines 80–86 could be moved to lines 62–63 for better logical flow and clearer connection to subsequent hypotheses. The logical progression between the importance of studying the effects of spatial representations of time on object-based attention and the stated objectives of the study is unclear. This connection is revisited later with the study goals but remains ambiguous. Additionally, there is no clear explanation of IOR in terms useful for understanding the subsequent results, making it difficult to follow the authors’ discussion. The hypotheses are not clearly articulated in terms of computational models and cognitive processes, nor are they clearly linked to the experimental protocol described above. Specifically, hypothesis 2 could potentially be explained as a priming effect, which raises questions about how the authors' innovative hypothesis differs from this explanation. |
||
Response 2: Thank you for your thoughtful and detailed feedback on the introduction. We have made substantial revisions to this section to improve its clarity and logical flow. Building on your suggestions, we reorganized the content in a way that best aligns with the overall structure and objectives of the manuscript. Specifically, we addressed redundancies and strengthened the connection between the theoretical background and the study’s objectives, ensuring a more coherent progression of ideas. Regarding the explanation of IOR, we decided to remove this content from the introduction, as it is more appropriately discussed in the context of our experimental findings. Instead, we have included detailed discussions of IOR in the discussion sections of each experiment and the general discussion, where it is directly relevant to interpreting our results. Finally, while the introduction does not explicitly address alternative explanations such as priming effects, we have provided a detailed clarification of the distinction between the priming effect and the effect of image similarity proposed in our hypothesis in our response to Comments 7. We believe these revisions have improved the clarity and coherence of the introduction. |
||
Comments 3: Experiment 1: Were all participants right-handed? Moreover, do the authors believe that the consistent gender imbalance across groups and experiments might have introduced any differences? Additionally, the main text does not state whether participants signed informed consent. |
||
Response 3: We thank you for raising these important points. We have carefully considered your comments and revised the manuscript accordingly. Below, we provide detailed responses to each of the concerns you raised. Handedness: The revised manuscript now includes detailed information on participant handedness in “Participants” sections. Experiment 1a included one left-handed participant; Experiment 1b included only right-handed participants; Experiment 2 included one left-handed participant; and Experiment 3 included two left-handed participants. Your comment also prompted us to further consider the potential influence of handedness on our findings. Previous research (Jones & Martin, 1997) suggested that right-handed individuals might perform worse than chance on recall tasks involving directional processing, unlike left-handed individuals. This potential suppression of past information among right-handed participants, given that recall tasks require retrieval of past-related information, could support our finding of the selective suppression of previously stored information presented in a "past"-metaphorical location. This discussion was elaborated in the General Discussion section of the revised manuscript (page 16, lines 686-695). Gender Imbalance: We appreciate your concern regarding the gender imbalance across our experiments. While we acknowledge this imbalance, which stemmed from the characteristics of our participant pool (predominantly female volunteers), we believe it is unlikely to have significantly influenced our results. For example, Qian et al. (2015) conducted three experiments with varying gender distributions (e.g., 5 females out of 16 participants in Experiment 1, 10 females out of 20 participants in Experiment 2, and 7 females out of 16 participants in Experiment 3). Despite these differences, the study did not report any significant gender effects on outcomes related to attentional bias or task performance. This suggests that gender was not a critical factor influencing the results. Similarly, evidence from other studies (e.g., Schneider et al., 2015) supports the conclusion that gender differences are unlikely to play a significant role in tasks involving spatial cueing paradigms or memory representation. Although we do not believe the gender imbalance impacted our findings, we recognize the importance of balanced gender representation in research. In future studies, we will aim to recruit a more gender-balanced sample to further enhance the generalizability of our results. Informed Consent: The revised manuscript now explicitly states that all participants provided informed consent prior to participation (page 3, lines 112). Jones, G.V.; Martin, M. Handedness Dependency in Recall from Everyday Memory. British Journal of Psychology 1997, 88, 609–619, doi:10.1111/j.2044-8295.1997.tb02660.x. Schneider, D.; Mertes, C.; Wascher, E. On the fate of non-cued mental representations in visuo-spatial working memory: Evidence by a retro-cuing paradigm. Behav. Brain Res. 2015, 293, 114–124. https://doi.org/10.1016/j.bbr.2015.07.034. Qian, Q.; Wang, F.; Feng, Y.; Song, M. Spatial organisation between targets and cues affects the sequence effect of symbolic cueing. J. Cogn. Psychol. 2015, 27, 752–764. https://doi.org/10.1080/20445911.2015.1048249. |
||
Comments 4: Experiment 1: It is unclear whether the disorder in the conditions (particularly 1/4D and 1/2AD) always pertained to the right-hand side of the image (as shown in Fig. 2) or also to the left-hand side. Clarifying this would help disambiguate whether the effect is due to greater attention directed to the disordered quarter of the image. Additionally, it is unclear whether the intent of the experiment was to induce a Posner effect using the images in different positions. If so, the congruence and incongruence percentages of the cue relative to the actual presentation should be specified. Furthermore, the rationale behind the selection of these specific images, which could themselves induce spatial attentional biases, is not explained. |
||
Response 4: Location of Disordered Regions: We appreciate your insightful observation regarding the importance of specifying the location of the disordered regions. To address this, we ensured that any observed effects were not influenced by spatial biases arising from fixed structural differences within the images. Specifically, within each disorder level, the location of the disordered regions varied systematically across images, ensuring that they were evenly distributed across different positions. For <1/4D and 1/4D images, disorder was equally distributed among the four quadrants. Similarly, for images with 1/2AD disorder, the disordered regions were placed in different orientations (top, bottom, left, right) across the stimuli set, while for 1/2ND images, disordered regions were balanced across every pair of non-adjacent quadrants. This balanced approach, combined with randomization, was designed to minimize any potential bias stemming from a fixed spatial preference for the disordered portions of the images. We have expanded on this aspect of our stimulus design in the "Apparatus and Stimuli" section of Experiment 1 in the revised manuscript to clarify how the balanced distribution of disordered regions was achieved across different conditions. This information is now detailed on page 3, lines 130–135. Posner Effect and Cue Validity: Thank you for raising this important point. The aim of the study was indeed to investigate attention biases by examining differences in the Posner effect. To ensure a rigorous and unbiased test, we adopted a balanced design, with 50% of the trials featuring valid cues (where the cue and target appeared in the same location, congruent trials) and 50% featuring invalid cues (where the cue and target appeared in different locations, incongruent trials). This clarification has been included in the revised "Design and Procedure" section of Experiment 1 (page 4, lines 153–154). Rationale for Image Selection: Thank you for your comment. We have updated the manuscript to provide further details on the image selection process. Specifically, we controlled for luminance, size, shape, and color to ensure consistency and minimize potential attention biases across all experimental conditions. Furthermore, we systematically manipulated five levels of disorder to ensure that any observed effects could be attributed to the experimental manipulations rather than pre-existing biases in the stimuli. These updates have been incorporated into the 'Apparatus and Stimuli' section and can be found on page 3, lines 127–135. |
||
Comments 5: Experiment 1: Regarding the results, the significance of the discrepancy score used to demonstrate the rightward effect of future events is not adequately explained. Merely defining how the score is calculated does not help integrate the information and understand the results. This applies throughout the manuscript whenever the term “discrepancy” or “difference” is used to describe results. If the significant differences reported in the figures pertain to the discrepancy score, it would be better to explicitly state this for easier interpretation. |
||
Response 5: In the revised "Results for RTs in Experiment 1a" section (pages 4–5, lines 180–190), we now explicitly detail how the discrepancy score of Image Position is calculated and its relevance to our findings. Specifically, this score is calculated as the difference in reaction times (RTs) for images displayed on the left versus the right (RT left minus RT right). A positive discrepancy score indicates a rightward attentional bias, while a negative score indicates a leftward bias. The magnitude of the discrepancy score reflects the strength of this attentional bias, with larger values indicating stronger asymmetries in attention allocation between the left and right visual fields. This measure is directly tied to our hypothesis that individuals would exhibit a rightward attentional bias in the horizontal orientation, consistent with the future-oriented spatial metaphor. By using the discrepancy score to quantify this bias, we were able to directly test our hypothesis and statistically evaluate the strength of the predicted effect. Additionally, in Experiment 2, we have provided further explanation of the discrepancy score of Image Similarity (page 8, lines 322-330), which captures individual differences in responses to cue images that either match or differ from the central image. For RTs, a positive discrepancy score indicates faster responses to matching images, while a negative score indicates slower responses to matching images. For error rates, a positive score reflects fewer errors for matching images, whereas a negative score reflects more errors for matching images. These clarifications have been incorporated in the revised manuscript to ensure that the presentation of our results is clearer and more accessible. |
||
Comments 6: Experiment 1: In the discussion of Experiment 1, it is unclear how the rightward attentional bias is attributed to the future-oriented spatial representation of time and not to some other task-related effect. It is also unclear why this conclusion is drawn specifically from the observation that the image with disorder in the 1/4D condition induces an association between the right position and the future. The connection between a Posner paradigm and the spatial representation of time, which remains implicit in the task, is not made clear. |
||
Response 6: In Experiments 1a and 1b, we employed a spatial cueing paradigm (Posner paradigm) designed to implicitly activate associations between spatial positions and temporal concepts by presenting stimuli in horizontal or vertical orientations, without explicitly instructing participants to focus on temporal concepts. This paradigm minimized direct, goal-directed attention to the stimuli or their positions, as participants' primary task was to respond to the probe dot independently of the stimulus presentation. By decoupling the primary task from the spatial and temporal properties of the stimuli, this design reduces the likelihood that task-specific effects, such as direct attention to stimuli or their positions, could influence the results. Thus, differences in reaction times and accuracy across stimulus positions are more likely to reflect participants’ ingrained spatial-temporal attentional biases rather than task-related factors. Moreover, the Posner paradigm inherently leverages spatial cueing mechanisms to activate spatial-temporal associations implicitly, as participants process the spatial arrangement of stimuli (e.g., horizontal orientation) in a manner consistent with the future-oriented spatial metaphor. This implicit activation allows us to assess attentional biases linked to spatial-temporal representations without explicitly instructing participants to consider temporal concepts. The observed rightward attentional bias under higher disorder levels (1/2AD and 1/2ND conditions) aligns with the findings on error rates and supports the future-oriented spatial metaphor, wherein participants from left-to-right reading cultures associate the right side with the "future" and the left side with the "past." This interpretation is consistent with prior research on spatial-temporal associations. In contrast, the leftward bias observed in the 1/4D condition deviates from this pattern, suggesting that intermediate levels of disorder may impose unique attentional demands or cognitive loads, which could temporarily disrupt or override default spatial-temporal associations. This result indicates that attentional allocation may be sensitive to specific levels of visual order/disorder, leading to deviations from typical spatial-temporal biases. However, the mechanisms underlying this effect remain unclear, as no prior research has systematically examined how varying levels of disorder interact with spatial-temporal associations. Despite this deviation, the consistent rightward bias observed in the 1/2AD and 1/2ND conditions supports the conclusion that attentional biases are influenced by future-oriented spatial metaphors. Further research is needed to systematically investigate how varying levels of disorder interact with attentional allocation and spatial-temporal associations, potentially uncovering new insights into the dynamic nature of these biases. |
||
Comments 7: Experiment 2: What is the difference between the effect of image similarity and the priming effect? Additionally, the explanation of the IOR observed in the study is not well contextualized within the experiment. |
||
Response 7: Thank you for your insightful comment. We have addressed the two points as follows: Difference between Image Similarity Effect and Priming Effect: In our study, the image similarity effect refers to the faster and more accurate responses observed when participants encountered cue images that were completely identical to the central image. While this effect can be considered a form of priming, which involves the influence of one stimulus on the processing of a subsequent stimulus, the key distinction lies in its specificity. Priming is a broader concept that encompasses various cognitive processes and often occurs without conscious awareness. In contrast, the image similarity effect we focus on is explicitly tied to the visual matching of cues and target stimuli based on their identical features. In our experimental design, image similarity was operationalized as a direct and controlled comparison between "same" (completely identical) and "different" conditions, explicitly testing visual matching effects rather than the implicit cognitive mechanisms underlying priming. While there are conceptual overlaps, we interpret our findings within the more specific framework of image similarity effects to align with our emphasis on visual processing in spatial attention. Contextualization of Inhibition of Return (IOR): Regarding the explanation of IOR, we have revised the manuscript to better integrate this phenomenon with the experimental findings. Specifically, we removed the discussion of IOR from the introduction, as it was not a predicted outcome of our experimental design. Instead, we have elaborated on IOR in the discussion sections of each experiment and the general discussion, where it is directly relevant to the interpretation of our results. For example, in Experiment 1, we considered the potential presence of IOR in light of the non-significant main effect of cue validity. In Experiments 2 and 3, significant IOR effects were observed, with participants responding more slowly and less accurately to valid cues. These findings suggest that memory representations not only facilitated attentional engagement but also contributed to the amplification of IOR. As discussed in detail in the discussion sections of each experiment and the general discussion, we have situated the explanation of IOR within the context of the experimental results to provide a more precise and focused interpretation of its role in our study. We kindly invite you to refer to these sections for a more comprehensive discussion of this phenomenon. |
||
Comments 8: Experiment 3: Line 434 contains a typo: it refers to error rates but seems to discuss reaction times (RT). |
||
Response 8: Thank you for pointing out the typo. We have corrected it in the revised manuscript to accurately refer to reaction times (RT) instead of error rates. This update can be found on page 12, line 502 in the revised version. We appreciate your careful review. |
||
Comments 9: General Discussion: Lines 497–499: Based on the study’s results, how can the facilitation in responding to images presented on the right be conclusively attributed to their association with the future? While the theoretical foundation of the spatial representation of time is compelling, it is unclear whether the task employed truly supports this hypothesis. Moreover, the discussion would benefit from being expanded to include additional literature supporting the results provided by the study. |
||
Response 9: Thank you for your constructive comments. We have revised the General Discussion to clarify how the observed rightward bias can be attributed to future-oriented processing. The spatial cueing paradigm in Experiment 1 was specifically designed to minimize potential confounds, such as memory interference or task-related factors, allowing us to isolate intrinsic spatial-temporal associations. This provides robust evidence that the rightward bias stems from these associations rather than external influences, diverging from prior studies that focused on past-oriented attentional biases driven by memory (e.g., Awh et al., 2006; Downing, 2000; Kruijne & Meeter, 2016; Oberauer, 2019). To further understand the mechanisms underlying the future-oriented attentional bias found in our study, we also considered the potential contribution of hemispheric processing biases. While previous research (e.g., Geffen et al., 1971) has shown that stimuli in the right visual field (RVF) are processed more efficiently due to the left hemisphere's specialization for verbal processing, our systematic manipulation of disorder levels revealed attentional shifts that cannot be fully explained by hemispheric asymmetries. For instance, the reversal of the rightward bias in the 1/4D condition suggests that verbal processing plays a limited role in shaping these effects. These findings highlight the dominant role of spatial-temporal associations in driving the future-oriented bias. Furthermore, the less pronounced but consistent downward bias observed in Experiment 1b suggests that future-oriented spatial biases extend beyond the horizontal plane, reflecting a broader attentional mechanism. Additionally, recent work by Liu et al. (2024) provides strong support for our findings, demonstrating a gradual shift in attention from past to future through participants’ eye movements. This aligns with our results and reinforces the idea that spatial-temporal associations are deeply embedded in attentional mechanisms. These revisions aim to provide a more comprehensive theoretical framework for interpreting the observed biases and their functional implications. We believe this expanded discussion strengthens the manuscript by situating our findings within the broader context of spatial-temporal research and attentional processing. Please refer to page 14, lines 574–599 for the updated content. We hope these revisions address your concerns and provide a clearer and more compelling interpretation of our findings. Awh, E.; Vogel, E.K.; Oh, S.-H. Interactions between Attention and Working Memory. Neuroscience 2006, 139, 201–208, doi:10.1016/j.neuroscience.2005.08.023. Downing, P.E. Interactions between Visual Working Memory and Selective Attention. Psychol Sci 2000, 11, 467–473, doi:10.1111/1467-9280.00290. Kruijne, W.; Meeter, M. Implicit Short- and Long-Term Memory Direct Our Gaze in Visual Search. Atten Percept Psychophys 2016, 78, 761–773, doi:10.3758/s13414-015-1021-3. Oberauer, K. Working Memory and Attention - A Conceptual Analysis and Review. J Cogn 2019, 2, 36, doi:10.5334/joc.58. Geffen, G.; Bradshaw, J.L.; Wallace, G. Interhemispheric Effects on Reaction Time to Verbal and Nonverbal Visual Stimuli. Journal of Experimental Psychology 1971, 87, 415–422, doi:10.1037/h0030525. Liu, B.; Alexopoulou, Z.-S.; van Ede, F. Jointly Looking to the Past and the Future in Visual Working Memory. eLife 2024, 12, RP90874, doi:10.7554/eLife.90874. |
||
4. Response to Comments on the Quality of English Language |
||
Comment: The quality of English does not limit my understanding of the research. |
||
Response: Thank you, we'll continue to improve our English to better express our research! |
||
5. Additional clarifications |
||
Thank you again for your valuable and thorough feedback, which has greatly improved the quality of our manuscript! We look forward to hearing your thoughts on the revised version and sincerely appreciate your continued guidance! |
Reviewer 2 Report
Comments and Suggestions for Authors
The authors present four behavioral spatial cueing experiments in which visual patterns with different levels of disorder served as exogenous cues. In Experiments 2 and 3, the task included an additional short-term memory component as cue images could be identical to centrally presented initial images. With these experiments, the authors aimed to examine the effects of the spatial metaphor of time, of short-time memory representations and of Gestalt-aligned order on visual attention – a rather comprehensive undertaking that succeeds only partly.
In line with the authors’ ambitious research question, the experiments presented here feature complex three-factor designs with up to five levels per factor. These complex designs yield similarly complex result patterns, yet only a select few of the results are further discussed. Unfortunately, most of these discussed results have already been found in previous research: Experiments 1a and 1b show lower error rates for cue stimuli presented on the right as compared to the left and faster RTs for those presented at the bottom as compared to the top, respectively. The authors claim that this indicates an attentional bias towards right (and bottom) as future-related spatial locations, but the results could just as well be explained by hemispheric processing biases: stimuli that can be verbally encoded, such as the dot patterns used in the present study, are processed more quickly when presented in the right visual field (e.g. Geffen et al., 1971).
Experiment 2 showed a cue validity effect consistent with inhibition of return and that cue images which were identical to previously presented central images induced faster and more accurate probe reactions. As the authors themselves note “this aligns with previous research” (l. 334 f.).
Experiment 3 again found inhibition of return and (in contrast to Experiment 2) a null effect of Image Similarity that the authors attribute to the fact that cue images that were not identical to the preceding central image were always fully-ordered patterns and thus conforming to Gestalt-principles. A more convincing argument for the effect of Gestalt-principles on attention could have been made if the Image Disorder manipulation that sets this study apart from others had more of an effect on results. This was not the case and what little effect the manipulation did have is not sufficiently discussed (see the Image Position x Disorder Level interaction mentioned below).
As such, I am unsure whether the present study adds anything substantial to the literature and cannot recommend its publication. Below, I have listed the main issues I have with the manuscript beyond its lack of relevance.
The methods sections of the various experiments omit crucial information, e.g.: what was the proportion of valid to invalid trials? How many trials were available at each level of the experimental design? Why was cue image presentation time variable, what was the proportion of 100ms presentation to 200 ms presentation durations, was this proportion the same in all conditions and did cue presentation duration have any effect on the results? Did participants respond with both hands? If so, due to the lack of response key randomization, any RT differences between left/top (F-key/T-key) and right/bottom (J-key/N-key) responses might simply result from the latter being performed with the right hand which was presumably the dominant hand in the majority of participants.
The result sections of all experiments should include a figure or table displaying mean RT and error rates across all conditions. In l. 181 f., the authors indicate that data from Experiments 1a and 1b were previously used in separate analyses and are considered for publication elsewhere. The authors need to characterize what kind of analyses these were, especially because the present manuscript also features separate analysis of the data.
The discussion for each individual experiment is much too concise. For one, the authors fail to follow up each experiment with a rationale for why the subsequent experiment should be conducted. More importantly, both in the individual and the general discussion, the authors make wide-reaching claims, but never indicate which of their results in particular support these claims (e.g. l.212-214: “The results of Experiments 1a and 1b indicated that participants displayed an attentional bias toward the right side in the horizontal orientation, supporting the future-oriented spatial metaphor. In the vertical orientation, attention biased downward, though this effect was less pronounced.“) As noted above, the majority of results is not discussed at all. For example, in Experiment 1, the authors found a significant Image Position x Disorder Level interaction, that was due to cue images with high disorder levels (and to a lesser degree also those with low disorder) leading to faster reaction times for right probes. Only cue images with one-quarter disorder produced slightly faster reaction times for left cues. The authors deem this interaction important enough to illustrate it in Figure 3, but then never mention it again in the manuscript. Similarly, the authors fail to discuss unexpected null results, such as the lack of cue validity effects in Experiment 1.
In Experiments 2 and 3, any effects involving the factor Image Position (left/right/top/bottom) could potentially be an artefact of the fixed task order, in which participants always first completed the task on stimuli presented on the left/right first and second on those presented at the top/bottom. In keeping with this, across both experiments, reaction times are consistently faster for left/right stimuli compared to top/bottom stimuli.
Geffen, G., Bradshaw, J. L., & Wallace, G. (1971). Interhemispheric effects on reaction time to verbal and nonverbal visual stimuli. Journal of experimental psychology, 87(3), 415.
Author Response
Response to Reviewer 2 Comments
|
||
1. Summary |
|
|
The authors present four behavioral spatial cueing experiments in which visual patterns with different levels of disorder served as exogenous cues. In Experiments 2 and 3, the task included an additional short-term memory component as cue images could be identical to centrally presented initial images. With these experiments, the authors aimed to examine the effects of the spatial metaphor of time, of short-time memory representations and of Gestalt-aligned order on visual attention – a rather comprehensive undertaking that succeeds only partly. Response: Thank you very much for your comprehensive review and the insightful comments provided. We deeply appreciate your expertise and have carefully considered your feedback in full. This includes areas where your thoughtful recommendations have significantly refined our approach, as well as points that we believe could benefit from further discussion and clarification. Below, we provide a detailed explanation of the modifications made to the manuscript in response to your comments. |
||
2. Questions for General Evaluation |
Reviewer’s Evaluation |
Response and Revisions |
Is the content succinctly described and contextualized with respect to previous and present theoretical background and empirical research (if applicable) on the topic? |
Must be improved |
We genuinely value the insightful feedback provided by the reviewers. In light of your comments, we have made revisions to the introduction, methods, and discussion sections to better articulate the theoretical background and experimental hypotheses. The research methodology has been elaborated in detail, and the discussion has been carefully refined to ensure coherence, balance, and alignment with the results and relevant literature. These revisions aim to enhance the clarity, rigor, and overall quality of the manuscript.
|
Are the research design, questions, hypotheses and methods clearly stated? |
Must be improved |
|
Are the arguments and discussion of findings coherent, balanced and compelling? |
Must be improved |
|
For empirical research, are the results clearly presented? |
Must be improved |
|
Is the article adequately referenced? |
Can be improved |
|
Are the conclusions thoroughly supported by the results presented in the article or referenced in secondary literature? |
Must be improved |
|
3. Point-by-point response to Comments and Suggestions for Authors |
||
Comments 1: In line with the authors’ ambitious research question, the experiments presented here feature complex three-factor designs with up to five levels per factor. These complex designs yield similarly complex result patterns, yet only a select few of the results are further discussed. Unfortunately, most of these discussed results have already been found in previous research: Experiments 1a and 1b show lower error rates for cue stimuli presented on the right as compared to the left and faster RTs for those presented at the bottom as compared to the top, respectively. The authors claim that this indicates an attentional bias towards right (and bottom) as future-related spatial locations, but the results could just as well be explained by hemispheric processing biases: stimuli that can be verbally encoded, such as the dot patterns used in the present study, are processed more quickly when presented in the right visual field (e.g. Geffen et al., 1971). Experiment 2 showed a cue validity effect consistent with inhibition of return and that cue images which were identical to previously presented central images induced faster and more accurate probe reactions. As the authors themselves note “this aligns with previous research” (l. 334 f.). Experiment 3 again found inhibition of return and (in contrast to Experiment 2) a null effect of Image Similarity that the authors attribute to the fact that cue images that were not identical to the preceding central image were always fully-ordered patterns and thus conforming to Gestalt-principles. A more convincing argument for the effect of Gestalt-principles on attention could have been made if the Image Disorder manipulation that sets this study apart from others had more of an effect on results. This was not the case and what little effect the manipulation did have is not sufficiently discussed (see the Image Position x Disorder Level interaction mentioned below). As such, I am unsure whether the present study adds anything substantial to the literature. Geffen, G., Bradshaw, J. L., & Wallace, G. (1971). Interhemispheric effects on reaction time to verbal and nonverbal visual stimuli. Journal of experimental psychology, 87(3), 415. |
||
Response 1: We sincerely thank you for your thoughtful and detailed feedback regarding the complexity of our experimental designs and their implications for the interpretation of our findings. We appreciate the opportunity to address the concerns raised and provide clarification on the contributions of our study. Below, we respond to the specific points raised: On the complexity of the experimental designs and discussed results: We acknowledge that the three-factor designs in our experiments are indeed complex, and this complexity inevitably results in intricate patterns of findings. In our discussion, we chose to focus on the most salient and theoretically relevant results rather than addressing all observed patterns. This decision was made to ensure that the discussion remains aligned with the primary objectives of our study and avoids unnecessary overextension. While the findings from Experiment 1 may appear less novel, they serve as a critical foundation for the subsequent experiments (Experiments 2 and 3). Specifically, Experiment 1 was designed to establish baseline attentional biases in response to varying levels of disorder, which are essential for interpreting the additional factor of short-term memory representations introduced in Experiments 2 and 3. The results of Experiment 1 thus provide a necessary reference point for understanding how attentional biases interact with memory processes in the later experiments. By situating Experiment 1 as a foundational step, we aimed to build a coherent progression in our research design, where each experiment contributes incrementally to addressing the overarching research question. We believe this approach strengthens the interpretability and theoretical significance of the findings across all three experiments. Additionally, we appreciate your suggestion to consider hemispheric processing biases, as referenced in Geffen et al. (1971). We agree that hemispheric processing biases, particularly for stimuli that can be verbally encoded, could offer an alternative explanation for the findings observed in Experiment 1. However, based on a thorough analysis of the results, it appears that the present work extends beyond a strictly hemispheric bias interpretation: â‘ Stimulus Control and Balanced Design We used OCTA-generated images at various levels of disorder, rather than purely linguistic or symbolic content, such as letters or numbers. Although the dot patterns may have some degree of verbalizability, we took care to balance luminance, color, and shape, and randomized image positions across trials. This design was intended to minimize consistent visual field advantages driven solely by verbal encoding. â‘¡Temporal-Spatial Associations Our primary research question was whether individuals from a left-to-right reading culture might implicitly map the future onto the right side. If purely hemispheric bias were driving our effects, we would expect similar rightward trends across all disorder levels or in tasks without any spatial dimension. Interestingly, however, we observed a shift from leftward to rightward biases at certain disorder levels. For example, discrepancy score analysis of RTs showed a rightward attentional bias under higher disorder levels (1/2AD and 1/2ND), contrasting with a leftward bias in the 1/4D condition. This shift suggests that attentional allocation may not be entirely explained by hemispheric lateralization. Furthermore, in the vertical orientation, we observed a less pronounced but still noticeable tendency for faster reactions to the bottom. Previous research indicates that humans have an inherent preference for the lower visual field, which may exist independently of hemispheric dominance. This suggests that something beyond left versus right hemisphere processing is influencing the attentional shifts we observed, including widely reported spatial-temporal metaphors. Thanks again for this comment and we have incorporated a more detailed discussion of the potential contribution of hemispheric biases in the revised manuscript (see page 14, lines 584–596). In future research, we plan to incorporate additional controls, such as lateralized presentation of verbal and nonverbal stimuli or the use of eyetracking measures, to further disentangle hemispheric processing from spatial-temporal mappings. On the contribution of the study: â‘ Cue Validity and Inhibition of Return (IOR): While the cue validity effects observed in both Experiments 2 and 3 are consistent with inhibition of return (IOR), aligning with previous research, our study provides new insights into IOR and its relationship with short-term memory representations. Specifically, unlike Experiments 2 and 3, Experiment 1 did not reveal a significant main effect of cue validity. This finding suggests that short-term memory representations may play a critical role in suppressing previously processed information and triggering IOR. Although IOR itself is not a novel phenomenon, the design of our three experiments builds upon this well-established effect to deepen our understanding of the mechanisms underlying attentional processes. By systematically layering the manipulations across the experiments, our study highlights how IOR interacts with short-term memory representations, offering a more nuanced perspective on how attention is allocated and inhibited in dynamic visual environments. â‘¡The finding and discussion of Image Disorder Thank you for pointing out this important aspect of our study. You raise an excellent point regarding the role of Image Disorder manipulation and its potential to provide deeper insights into the effects of visual order on attention. To be candid, our initial goal during the experimental design phase was indeed to explore how varying levels of visual disorder might influence attentional processes. However, it is possible that the simplicity of both the stimulus materials and the response tasks limited our ability to capture more fine-grained effects through reaction times and error rates. As discussed in greater detail in the General Discussion section (pages 16, lines 696–720), we acknowledge that the subtle effects of Image Disorder require further exploration and that more sophisticated experimental designs may be necessary to fully unpack their influence on cognitive mechanisms such as attention and memory. We also recognize that the Image Position x Disorder Level interaction is an intriguing result, and we made every effort to consult the existing literature to provide a more thorough interpretation. Unfortunately, current research does not offer sufficient or convincing explanations for the differential effects observed across disorder levels. This highlights a gap in the literature that we hope to address in future studies. Specifically, we aim to design more sophisticated stimulus materials and adopt experimental paradigms better suited to capturing the nuanced effects of order and disorder on cognitive mechanisms such as attention and memory. Nevertheless, we believe our interpretation of the null effect of Image Similarity in Experiment 3 remains reasonable. As you noted, Experiment 3 replicated the IOR effect observed in Experiment 2, but the absence of an Image Similarity effect in Experiment 3 can be attributed to the fact that the cue images, while not identical to the preceding central image, were fully-ordered patterns conforming to Gestalt principles. The Gestalt principle likely reduced the distinctiveness of the cue images relative to the central image, thereby diminishing the influence of short-term memory represtation on attentional processes. This interpretation is supported by two key factors. First, our experimental design carefully controlled for extraneous variables, ensuring that the observed effects were not confounded by other factors. Second, the consistency of the IOR effect across Experiments 2 and 3 reinforces the robustness of our findings, while the differences in Image Similarity effects highlight the potential role of Gestalt principles in modulating attention. In summary, while our study has limitations in fully capturing the effects of Image Disorder, it nonetheless offers valuable insights into the interplay between visual order, Gestalt principles, and attentional mechanisms. Although the observed effects may appear subtle, they highlight, to some extent, the nuanced ways in which visual structure influences attention. We appreciate your thoughtful feedback, which has helped us reflect on these aspects and will guide the design of future studies to better address these questions! |
||
Comments 2: Below, I have listed the main issues I have with the manuscript beyond its lack of relevance. The methods sections of the various experiments omit crucial information, e.g.: what was the proportion of valid to invalid trials? How many trials were available at each level of the experimental design? Why was cue image presentation time variable, what was the proportion of 100ms presentation to 200 ms presentation durations, was this proportion the same in all conditions and did cue presentation duration have any effect on the results? Did participants respond with both hands? If so, due to the lack of response key randomization, any RT differences between left/top (F-key/T-key) and right/bottom (J-key/N-key) responses might simply result from the latter being performed with the right hand which was presumably the dominant hand in the majority of participants. |
||
Response 2: Thank you for your insightful comments and for raising these important points. Below, we provide detailed responses to address the specific questions: â‘ Proportion of valid to invalid trials: The proportion of valid and invalid trials was 50% each. This balanced design was implemented to ensure equal representation of both trial types, minimizing potential biases in participants' responses. We have included this information in the revised manuscript on page 4, lines 153–154. â‘¡The cue image presentation time was varied (100ms or 200ms) to prevent participants from forming fixed expectations about the duration of the cue. Fixed durations could lead to habituation, reducing attentional engagement and potentially influencing task performance. By randomizing the presentation durations (50% for each duration), we aimed to maintain participants' attention throughout the experiment, as suggested by findings in similar paradigms (Koppe et al., 2014). The 100ms and 200ms cue durations were presented in equal proportions (50% each) across all conditions and experiments. This consistent approach ensured that any observed effects were not confounded by unequal weighting of the presentation durations. We have added this information to the revised manuscript on page 4, lines 150–151. â‘¢Response method: Participants responded using both hands, which is standard practice in similar experimental paradigms. However, your comment prompted us to reflect further on the potential influence of handedness on our findings. While we acknowledge that the lack of response key randomization may introduce a potential confound related to hand dominance, previous research (Jones & Martin, 1997) provides an interesting perspective on this issue. Specifically, their findings suggest that right-handed individuals may perform worse than chance on recall tasks involving directional processing, in contrast to left-handed individuals. This aligns with our observation of selective suppression of previously stored information presented in a "past"-metaphorical location. The suppression of past-related information among right-handed participants is consistent with the spatial-temporal metaphor of time, where the left side is associated with the "past." We have incorporated this discussion into the revised manuscript (General Discussion, page 16, lines 686–695), as it provides a potential explanation for our findings and highlights an interesting avenue for future research. Koppe, G.; Gruppe, H.; Sammer, G.; Gallhofer, B.; Kirsch, P. Temporal unpredictability of a stimulus sequence affects brain activation differently depending on cognitive task demands. NeuroImage 2014, 101, 236–244. DOI: 10.1016/j.neuroimage.2014.07.008. Jones, G.V.; Martin, M. Handedness Dependency in Recall from Everyday Memory. British Journal of Psychology 1997, 88, 609–619, doi:10.1111/j.2044-8295.1997.tb02660.x. |
||
Comments 3: The result sections of all experiments should include a figure or table displaying mean RT and error rates across all conditions. In l. 181 f., the authors indicate that data from Experiments 1a and 1b were previously used in separate analyses and are considered for publication elsewhere. The authors need to characterize what kind of analyses these were, especially because the present manuscript also features separate analysis of the data. |
||
Response 3: Thank you for your feedback regarding the presentation of our results. Below, we address the specific points you raised: Presentation of mean RT and error rates across all conditions: While including comprehensive tables or figures for all conditions might aid understanding, due to the focus of our research and space constraints, we have chosen to present only those results that are directly relevant to our hypotheses in the main manuscript. However, we recognize the importance of providing a complete overview of the data and plan to include mean RTs and error rates across all conditions in the supplementary materials in future revisions. Moreover, to ensure transparency and accessibility, the stimuli and experimental data, including both raw and cleaned data, are already available on the Open Science Framework (OSF) at doi: 10.17605/OSF.IO/BDZ5K. This approach ensures that all relevant information is accessible while maintaining the clarity and focus of the main manuscript. Use of data from Experiments 1a and 1b in separate analyses: We appreciate the need for clarity regarding the use of data from Experiments 1a and 1b. These data were previously used in separate analyses focusing on the variation in attentional bias across varying levels of visual disorder, which are distinct from the present manuscript's focus on the impact of the spatial metaphor of time on attentional bias. We realize that mentioning these analyses without providing details may lead to confusion, especially since they have not yet been published. To avoid any misunderstanding and maintain the integrity of the current manuscript, we have opted to remove this statement from the text. We thank you for bringing this to our attention and agree that clear communication about the origin and use of data is essential for the transparency and credibility of our research. |
||
Comments 4: The discussion for each individual experiment is much too concise. For one, the authors fail to follow up each experiment with a rationale for why the subsequent experiment should be conducted. More importantly, both in the individual and the general discussion, the authors make wide-reaching claims, but never indicate which of their results in particular support these claims (e.g. l.212-214: “The results of Experiments 1a and 1b indicated that participants displayed an attentional bias toward the right side in the horizontal orientation, supporting the future-oriented spatial metaphor. In the vertical orientation, attention biased downward, though this effect was less pronounced.“) As noted above, the majority of results is not discussed at all. For example, in Experiment 1, the authors found a significant Image Position x Disorder Level interaction, that was due to cue images with high disorder levels (and to a lesser degree also those with low disorder) leading to faster reaction times for right probes. Only cue images with one-quarter disorder produced slightly faster reaction times for left cues. The authors deem this interaction important enough to illustrate it in Figure 3, but then never mention it again in the manuscript. Similarly, the authors fail to discuss unexpected null results, such as the lack of cue validity effects in Experiment 1. |
||
Response 4: Thank you for the thoughtful suggestions. We agree that the discussion sections for the individual experiments needed to be expanded to provide clearer rationales for the subsequent experiments and to better address the results. In response to your comments, we have thoroughly revised the manuscript. We have expanded the discussion sections for each experiment with additional references to provide more context and better clarify our findings. Particularly, we appreciate your observation regarding the lack of a significant cue validity main effect in Experiment 1. This was an important finding that prompted further reflection during our analysis, but we acknowledge that we had overlooked discussing it in the original manuscript. We have now addressed this in the discussion section (page 6, lines 238–253), providing potential explanations for this result and its implications. Regarding the significant Image Position × Disorder Level interaction, we recognize that this is an intriguing result that warrants further exploration. Unfortunately, the current body of research does not provide sufficient or convincing explanations for the differential effects observed across disorder levels. While we have made every effort to consult the existing literature and provide a thorough interpretation, we acknowledge that our understanding of this interaction remains limited. In fact, this result has inspired us to consider designing more sophisticated experiments in the future to better understand the nuanced effects of visual order/disorder on cognition such as attentional and memory processes. We are humbled by the complexity of this phenomenon and remain committed to addressing these questions in future research. We hope that our revisions and reflections demonstrate our commitment to providing a clearer and more comprehensive interpretation of our findings. |
||
Comments 5: In Experiments 2 and 3, any effects involving the factor Image Position (left/right/top/bottom) could potentially be an artefact of the fixed task order, in which participants always first completed the task on stimuli presented on the left/right first and second on those presented at the top/bottom. In keeping with this, across both experiments, reaction times are consistently faster for left/right stimuli compared to top/bottom stimuli. |
||
Response 5: Thank you for your insightful comment. This is indeed an important consideration in our study design. We confirm that Experiments 2 and 3 followed a fixed task order, with participants completing the horizontal task (left/right) first and the vertical task (top/bottom) second. This order was intentional and aligned with the objectives of our study. As explained in the introduction, prior research has shown that the spatial-temporal metaphor in the horizontal orientation is more universal and stable, while findings in the vertical orientation are less consistent. Therefore, our primary focus was on the horizontal orientation, with the vertical task included as a secondary exploration to provide additional context. To ensure the reliability of the horizontal results and avoid potential fatigue or practice effects impacting this key dimension, we prioritized the horizontal task. We acknowledge, however, that the fixed task order could potentially introduce order effects, such as faster reaction times for left/right stimuli due to participants completing the horizontal task first. This is a valid concern, and we have carefully considered its implications. Importantly, while the reaction time differences between horizontal and vertical orientations may reflect such order effects, we believe they do not undermine the primary conclusions of our study. Specifically, our key findings center on the attentional bias patterns within each orientation (e.g., left vs. right, top vs. bottom), rather than direct comparisons between horizontal and vertical orientations. These within-orientation effects are unlikely to be confounded by task order. We also recognize the limitations of this design and agree that a counterbalanced task order would provide a more robust test of these effects. This is an important point that we will address in future studies by adopting a fully counterbalanced design to rule out potential order effects. We hope this explanation clarifies the rationale behind our design and how we have interpreted the results in light of these considerations. Thank you again for raising this important point, which has helped us to further refine the discussion of our findings. |
||
4. Response to Comments on the Quality of English Language |
||
Comment: The quality of English does not limit my understanding of the research. |
||
Response: Thank you, we will continue to refine our English to more effectively convey our research! |
||
5. Additional clarifications |
||
We sincerely appreciate your insightful feedback. In the revised General Discussion (Pages 16, lines 696–720), we have expanded on both the contributions and limitations of our study, positioning it as a preliminary exploration while highlighting its potential to inform future research designs and stimulus development. Your thoughtful comments have been instrumental in refining these points, and we are deeply grateful for your guidance throughout this process. We look forward to hearing your thoughts on the revisions and remain open to any further suggestions you may have. |